

# Measurement report: The Fifth International Workshop on Ice Nucleation Phase 1 (FIN-01): Intercomparison of Single Particle Mass Spectrometers

Xiaoli Shen[1], David M. Bell[2,a], Hugh Coe[3,4], Naruki Hiranuma[5,b], Fabian Mahrt[6,c], Nicholas A. Marsden[3,4], Claudia Mohr[5,a,d], Daniel M. Murphy[7], Harald Saathoff[5], Johannes Schneider[8], Jacqueline Wilson[2,e], Maria A. Zawadowicz[9,f], Alla Zelenyuk[2], Paul J. DeMott[10], Ottmar Möhler[5], and Daniel J. Cziczo[1,9,11]

[1]Department of Earth, Atmospheric, and Planetary Sciences, Purdue University, West Lafayette, IN 47907, USA
[2]Atmospheric, Climate, and Earth Sciences Division, Pacific Northwest National Laboratory, Richland, WA 99354, USA
[3]Department of Earth and Environmental Sciences, University of Manchester, Manchester, M13 9PL, UK
[4]National Centre for Atmospheric Sciences, University of Manchester, Manchester, M13 0PL, UK
[5]Institute of Meteorology and Climate Research, Karlsruhe Institute of Technology, Eggenstein-Leopoldshafen, 76344, Germany
[6]Department of Environmental System Sciences, Institute for Atmospheric and Climate Science, ETH Zürich, Zurich, 8092, Switzerland
[7]Earth System Research Laboratory, National Oceanic and Atmospheric Administration, Boulder, CO 80305, USA
[8]Max Planck Institute for Chemistry, Mainz, 55128, Germany
[9]Department of Earth, Atmospheric, and Planetary Sciences, Massachusetts Institute of Technology, Cambridge, MA 02139, USA
[10]Department of Atmospheric Science, Colorado State University, Fort Collins, CO 80523-1371, USA
[11]Department of Civil and Environmental Engineering, Massachusetts Institute of Technology, Cambridge, MA 02139, USA
[a]now at: Laboratory of Atmospheric Chemistry, Paul Scherrer Institute, 5232 Villigen, Switzerland
[b]now at: Department of Life, Earth, and Environmental Sciences, West Texas A&M University, Canyon, TX 79016, USA
[c]now at: Department of Chemistry, Aarhus University, DK-8000 Aarhus, Denmark
[d]now at: Department of Environmental Systems Science, ETH Zürich, 8092 Zürich, Switzerland
[e]now at: UK Health Security Agency, Oak Park Lane, Cookridge, Leeds, West Yorkshire LS16 6RW, UK
[f]now at: Environmental and Climate Sciences Department, Brookhaven National Laboratory, Upton, NY 11973, USA

*Correspondence to*: Xiaoli Shen (xlshen@purdue.edu)

**Abstract.** Knowledge of chemical composition and mixing state of aerosols at a single particle level is critical for gaining insights into atmospheric processes. One common tool to make these measurements is single particle mass spectrometry. There remains a need to compare the performance of different single particle mass spectrometers (SPMSs). An intercomparison of SPMSs was conducted at the Aerosol Interaction and Dynamics in the Atmosphere (AIDA) chamber at the Karlsruhe Institute of Technology (KIT) in November 2014, as the first phase of the Fifth International Workshop on Ice Nucleation (FIN-01). In this paper we compare size distributions and mass spectra of atmospherically relevant particle types measured by five SPMSs. These include different minerals, desert and soil dusts, soot, bioaerosol (Snomax; protein granule), secondary organic aerosol (SOA) and SOA coated mineral particles. All SPMSs reported similar vacuum aerodynamic diameter ($d_{va}$) within typical instrumental ranges from ~100–200 nm (lower limit) to ~2–3 µm (upper limit). In general, all SPMSs exhibited a wide dynamic



range (up to $\sim10^3$) and high signal to noise ratio (up to $\sim10^4$) in mass spectra. Common spectral features with small diversities in mass spectra were found with high average Pearson's correlation coefficients, i.e., for average positive spectra $r_{avg-pos} = 0.74 \pm 0.12$ and average negative spectra $r_{avg-neg} = 0.67 \pm 0.22$. The highest correlation of average mass spectra across all instruments was observed for Snomax ($r_{avg-pos} = 0.92 \pm 0.04$ and $r_{avg-neg} = 0.90 \pm 0.05$), attributed to the prevalence of common markers and similar spectral patterns. The poorest correlation was found for propane soot ($r_{avg-pos} = 0.51 \pm 0.23$ and $r_{avg-neg} = 0.35 \pm 0.26$), primarily because of low detection efficiency (DE) due to small particle size. We found that instrument-specific DE was more dependent on particle size than particle type. We also found that particle identification favored the use of bipolar, rather than monopolar, instruments. Particle classification from "blind experiments" showed that all instruments differentiated SOA, soot, and soil dust, and detected subtle changes in the particle internal mixing, but had difficulties differentiating among specific mineral types and dusts. This study helps to further understand the capabilities and limitations of the single particle mass spectrometry technique in general, as well as the specific instrument performance in characterizing atmospheric aerosol particles. We propose that intercomparison workshops should continue as new SPMSs are developed and, ideally, should include both laboratory and field activities.

## 1 Introduction

Aerosol-cloud interactions are one of the largest uncertainties in the climate system (IPCC, 2021). A considerable source of uncertainty is related to an insufficient understanding of how chemical composition of aerosols affects their ability to act as cloud condensation nuclei (CCN) and ice nucleating particles (INPs). Both CCN and INPs influence cloud formation, microphysical and radiative properties, and precipitation formation and hence impact climate and the hydrological cycle (Lohmann et al., 2016). Studying INPs is particularly challenging, given that in the atmosphere only $\sim1$ in $\sim10^5$ particles act as an INP (Rogers et al., 1998). The low number concentration of INPs and the complex atmospheric aging processes that can affect the particle's ability to nucleate ice, including the acquisition of coatings and/or evaporation of components (Cziczo et al., 2009; Friedman et al., 2011; Möhler et al., 2008), can challenge current measurement techniques (Cziczo et al., 2003, 2017; Fuzzi et al., 2015; Kanji et al., 2017). Knowledge of chemical composition and mixing state on a single particle level is therefore critical to understand atmospheric processes and correctly predicting atmospheric impacts (Riemer et al., 2019).

In the past three decades in-situ and real-time single particle mass spectrometry has been widely used to characterize the size and composition of individual particles (Murphy, 2007). Corresponding studies have led to an improved understanding of internal and external mixing of ambient aerosol particles, as well as particle origin and chemical transformation (Noble and Prather, 2000; Pratt and Prather, 2012). The first airborne single particle chemical characterization of cirrus ice residues (IRs, the particle remaining after ice crystal sublimation) was in the 2001 CRYSTAL-FACE study (Cziczo et al., 2004). Since then, the single particle mass spectrometer (SPMS) has emerged as the powerful tool for assessing the chemical composition of ice-nucleating particles (INPs) and for direct measurements of IRs (Cornwell et al., 2019; Cziczo et al., 2003, 2006, 2009, 2013, 2017; Cziczo and Froyd, 2014; DeMott et al., 2003; Kamphus et al., 2010; Lacher et al., 2021; Lin et al., 2017; Pratt and



Prather, 2009; Roth et al., 2016; Sullivan et al., 2010). This was one of the major motivating factors for organizing the intercomparison of SPMSs within the framework of the Fifth International Ice Nucleation (FIN) Workshops, which sought to conduct comprehensive comparisons of instruments in both laboratory and field settings. Further information about the FIN workshops can be found in DeMott et al. (2011, 2018). The first phase (FIN-01), focused on intercomparing SPMSs, was conducted at the Aerosol Interaction and Dynamics in the Atmosphere (AIDA) chamber located at the Karlsruhe Institute of

Technology (KIT) in November 2014.

Since their inception, SPMS design has varied from instrument to instrument while also continuously undergoing improvements (Brands et al., 2011; Clemen et al., 2020; Cziczo et al., 2006; Dragoneas et al., 2022; Du et al., 2024; Erdmann et al., 2005; Gaie-Levrel et al., 2012; Gemayel et al., 2016; Hünig et al., 2022; Jacquot et al., 2024; Li et al., 2011; Marsden et al., 2016; Passig et al., 2020; Prather et al., 1994; Thomson et al., 2000; Trimborn et al., 2000; Zawadowicz et al., 2020;

Zelenyuk and Imre, 2005; Zelenyuk et al., 2009b, 2015). Unlike the Aerodyne Aerosol Mass Spectrometer (AMS), used for bulk measurements of non-refractory aerosol components (DeCarlo et al., 2006), no commercial instrument has dominated SPMS design. Design choices and technical details were reviewed by Murphy (2007). SPMSs have various characteristics, some of which are instrument specific, but instruments have some similar components, including an inlet system, a particle detection and sizing region, an ionization region, and one or two mass spectrometers. Here, we will briefly describe some of

these key features:

1) A critical orifice, capillary, or aerodynamic lens (ADL) inlet to transmit and focus particles into a narrow beam in a low pressure region (Davis, 1977; Murphy and Thomson, 1995; Liu et al., 1995).

2) A detection and sizing region using one or two continuous wave (CW) lasers, e.g., 532 nm Neodymium-doped Yttrium Aluminum Garnet (Nd:YAG) or 405 nm diodes. In this region, particle light scattering and the vacuum aerodynamic diameter

($d_{va}$) of individual particles are determined and recorded. Particles with $d_{va}$ from ~100 nm up to ~3 μm are generally detected. Detection is influenced by several factors, such as the particle size, shape, morphology, and optical properties of particles, the wavelength, power, and beam dimensions of the detection lasers, as well as the distance between two lasers (Sinha and Friedlander, 1985; Su et al., 2004; Gemayel et al., 2016).

3) An ion source region where one or two laser pulses provides laser desorption and ionization (LDI). This allows for

subsequent identification of most particulate components from volatile and semi-volatile to refractory. For one step LDI, a pulsed UV laser, e.g. 193 nm Argon Fluoride (ArF) excimer or 266 nm Nd:YAG laser, is triggered to ablate and ionize the particle after detection (McKeown et al., 1991; Prather et al., 1994). Two-step LDI often employs an infrared (IR) pulse (e.g., $CO_2$ laser) for desorption followed by a lower wavelength pulse for ionization (Cabalo et al., 2000; Zelenyuk et al., 1999, 2009b). Two-step LDI is commonly used to reduce ion fragmentation (Morrical et al., 1998; Zelenyuk et al., 2009a).

4) One or two mass spectrometers, most commonly time-of-flight (ToF), are used for measurement of ions (Murphy and Thomson, 1995; Gard et al., 1997).

There are limitations inherent to laser ablation and ionization of atmospheric particles with respect to reproducibility of mass spectra and quantitative assessment of single particle composition. The complexity of LDI mechanisms varies between





instruments due to ionization laser wavelength and laser power intensity at the point of ablation/ionization. Different SPMSs

may have differences in completeness of particle ablation, leading to diversity in spectral signatures (Murphy, 2007; Reilly et al., 2000; Reinard and Johnston, 2008; Thomson et al., 1997; Zenobi and Knochenmuss, 1998). Effort has been put in to refining particle identification but quantification of specific particle components remains a challenge (Allen et al., 2006, 2000; Bein et al., 2006; Fergenson et al., 2001; Froyd et al., 2019; Gallavardin et al., 2008; Gemayel et al., 2017; Gross et al., 2000; Gunsch et al., 2018; Hatch et al., 2014; Healy et al., 2013; Jeong et al., 2011; Köllner et al., 2017; Lu et al., 2018; Marsden et

al., 2018, 2019; May et al., 2018; Qin et al., 2006; Ramisetty et al., 2018; Shen et al., 2018, 2019; Wenzel et al., 2003; Zawadowicz et al., 2017, 2020; Zhou et al., 2016). Data analysis and classification methods used by different SPMSs can also differ substantially.

Intercomparisons of SPMSs and data analysis methods remain rare (Kamphus et al., 2010; Lacher et al., 2021; Middlebrook et al., 2003; Murphy et al., 2007). Middlebrook et al. (2003) reported a comparison of SPMSs, including the Particle Analysis

by Laser Mass Spectrometer (PALMS, National Oceanic and Atmospheric Administration, NOAA), Aerosol Time-of-Flight Mass Spectrometer (ATOFMS, University of California at Riverside), and Rapid Single-Particle Mass Spectrometer II (RSMS-II, University of Delaware) during the Atlanta Supersite Project in 1999. They found comparable particle classes with similar number fractions measured by the three SPMSs for the entire measurement period. Murphy et al. (2007) showed geographically broad distribution of lead in single particles, by comparing and combing the results from aircraft and ground-based

measurements of different SPMSs: PALMS, a commercial ATOFMS (Model 3800, TSI, USA), and RSMS. Kamphus et al. (2010) compared the Single Particle Laser-Ablation Time-of-Flight Mass Spectrometer (SPLAT, Max Planck Institute for Chemistry, MPIC) and a commercial ATOFMS at the Jungfraujoch research station and showed comparable results for IRs and droplet residuals. They also reported differences due to different ionization lasers and detection efficiencies for the two instruments. In a similar study, Lacher et al. (2021) showed comparable results of total aerosol composition from the Aircraft-

based Laser ABlation Aerosol MAss Spectrometer (ALABAMA, MPIC) and the Laser Ablation Aerosol Time-of-Flight Mass Spectrometer (LAAPTOF, AeroMegt GmbH, Germany) for ambient measurements at the Jungfraujoch.

One of the main goals of FIN-01 was to intercompare some of the major SPMSs used for atmospheric aerosol research. This included the custom-built instruments PALMS (Massachusetts Institute of Technology, MIT) (Cziczo et al., 2006), ALABAMA (MPIC) (Brands et al., 2011), and miniSPLAT (Pacific Northwest National Laboratory, PNNL) (Zelenyuk et al.,

2015) and the commercial instruments ATOFMS (TSI Model 3800, USA; from the Eidgenössische Technische Hochschule, ETH) (Gard et al., 1997; Prather et al., 1994) and LAAPTOF (AeroMegt GmbH, Germany; from University of Manchester, UoM) (Marsden et al., 2016).

The specific objectives of FIN-01 were:

1) Compare mass spectral signatures for key atmospheric particle types, including desert and soil dusts, soot and biological

particles, as well as particles with coatings.

2) Compare instrument performance and data analysis techniques in "blind experiments", where participants did not know the particle types being sampled.



3) Evaluation of the SPMSs' ability to measure the chemical composition of IRs.

This paper provides an overview of FIN-01. Experiments on particles with coatings by organics are discussed in more details

elsewhere (Bertozzi, 2022; Bertozzi et al., 2024).

## 2 Methods

More than ninety independent experiments were conducted during the FIN-01 workshop. Nine co-located SPMSs and ancillary aerosol characterization instrumentation were utilized. Here we focus on a subset, fourteen specific experiments, targeting the objectives listed above. Data were provided for five of the nine SPMSs deployed during FIN-01: ALABAMA, miniSPLAT,

PALMS and a commercial ATOFMS and LAAPTOF.

### 2.1 SPMSs

Table 1 summarizes the main components and performance parameters of the five SPMSs. Note that the values summarized in Table 1 and presented in this section are attributions to the current literature and were not independently verified during FIN-01. All instruments have been described in detail previously (Brands et al., 2011; Cziczo et al., 2006; Gard et al., 1997;

Gemayel et al., 2016; Marsden et al., 2016; Prather et al., 1994; Shen et al., 2018; Su et al., 2004; Zelenyuk et al., 2015). They are functionally similar, but some instrumental differences are noteworthy: All used an ADL, but not with the same inlet flowrate. Laser wavelength, laser power, focal spot size, and beam direction varied across the instruments. Laser differences can impact several measurement parameters:

1) Detectable particle size range. PALMS, LAAPTOF, and ATOFMS detected similar size range of ~100-200 nm to ~3 μm

$d_{va}$, ALABAMA detected a narrower range of ~200 nm to ~1 μm $d_{va}$, while miniSPLAT detected ~50 nm to ~1.4 μm $d_m$ (electrical mobility diameter).

2) Detection efficiency (DE) is defined as the ratio of the number of particles detected and/or sized to the total number of particles entering the inlet. All instruments exhibited size dependent-DE for spherical polystyrene latex (PSL) particles. Most instruments also exhibit a less pronounced size dependence on morphology. It is important to note that DE for miniSPLAT

has been defined differently from the other SPMSs. MiniSPLAT operates simultaneously in a "dual data acquisition mode" where the size distribution is determined at a rate up to several thousand particles per second, while single particle mass spectra, and corresponding $d_{va}$, are acquired at a rate similar to the other SPMSs, ~20 particles per second (Zelenyuk et al., 2015). For particle size measurements, miniSPLAT often uses a dilution stage to lower aerosol number density to reduce coincidence of multiple particles in the detection laser beams. The variable amount of dilution for FIN-01 was not reported and is therefore

not addressed in this paper. The DE of miniSPLAT, for comparison to the other instruments, can be referenced either to the ratio of total particles sized after the dilution stage or the ratio of the particles for which the laser was triggered. The former is a more direct comparison of DE to the other SPMSs, but both are presented here for clarity. Note that the other SPMSs run in a single acquisition mode which records single particle mass spectra and corresponding sizes. Most often, particles size



distributions are acquired by separate instruments running in parallel. This difference in methodology leads to differences in
terminology, which are described here when relevant.

3) Hit rate (HR) is most commonly defined as the ratio of the number of useful spectra generated to the number of detected particles. PALMS had a HR of ~95% without obvious size or shape dependency (Cziczo et al., 2006), largely due to the proximity of the final detection beam and the ablation and ionization location. Other instruments had larger distances between detection and ablation, often due to the time required between laser trigger and firing. ALABAMA exhibited a size-dependent
HR of ~1% and ~75% for PSL of 200 and 400 nm, respectively (Brands et al., 2011), ATOFMS exhibited ~57% for 90 nm and ~ 93% for 290 nm PSL (Su et al., 2004), and the UoM LAAPTOF had maximum HR of 70%. Note that miniSPLAT has defined HR differently than the other SPMSs, as the ratio of the number of usable mass spectra to the number of laser triggers (Zelenyuk et al., 2015). This definition is specific to miniSPLAT, due to the aforementioned dual data acquisition mode. In this paper the former, more general, definition of HR is used.

4) Mass spectra quality related parameters, e.g., mass resolution, dynamic range of the ion signals, and signal to noise ratio (SNR). The aforementioned size detection ranges and DE, as well as mass spectra obtained in FIN-01 are examined and compared in Section 3.1.

In order to reduce data complexity, so-called "clustering algorithms" have been used to classify / categorize particle mass spectra, i.e., groups of mass spectra that share spectral similarities. Examples of such clustering algorithms include k-means,
fuzzy c-means, ART-2a neural network, and hierarchical clustering (Gross et al., 2010; Murphy et al., 2003; Reitz et al., 2016; Zelenyuk et al., 2006, 2008b). The methods used to analyze data collected by the SPMSs deployed during FIN-01 are summarized in Table 1. Because the individual methods have been previously published, we refrain from a detailed description of the different analysis algorithms to instead focus on comparing the resulting grouped data from FIN-01.

## 2.2 Ancillary instruments

Additional instruments used during FIN-01 included multiple condensation particle counters with different size ranges (CPC, TSI, Model 3010, 3022, and 3025), a scanning mobility particle sizer (SMPS, TSI, Inc., Model 3081 differential mobility analyzer, DMA, and Model 3010 CPC) and an aerodynamic particle sizer (APS, TSI Inc., Model 3321) to measure the particle size distributions.

## 2.3 Experiments

Experiments were conducted using two chambers at KIT: the 84 m$^3$ AIDA chamber and the 3.7 m$^3$ stainless steel aerosol preparation and characterization (APC) chamber (Möhler et al., 2001, 2003; Saathoff et al., 2003). The aerosol particles and associated generation methods are summarized in Table 2. The placement of SPMSs during FIN-01 was used to minimize distance to the AIDA and APC chambers and is shown in Fig. 1. Specific details of the coating experiments can be found in Bertozzi (2022).




A subset of experiments was termed "blind experiments" with a goal of comparing results from the SPMSs where the aerosol composition was not known by the instrument teams a priori. The blind experiments were conducted under the direction of three referees, not associated with specific instrument teams. The referees added different particle types to the APC chamber and then collected results before disclosure of the chamber contents. The objective of the referees was to provide a range of

mass spectral signatures and particle sizes. After injection, the SPMSs sampled the (unknown) particle mixture directly from APC chamber. Groups then provided results to the referees without discussion with referees or other teams. As shown in Fig. 2, there were two blind experiment periods (P1 and P2), each with a duration of 2 to 3 h. The particle samples added at the start of P1 were: α-pinene SOA, Argentinian soil dust, and elemental carbon soot (GSG generator; hereafter graphite soot). Prior to P2, particle-free air was added to the chamber (i.e., dilution) and more graphite soot was added. The number percentage

of α-pinene SOA, Argentinian soil dust, and graphite soot were approximately 24%, 41%, and 35%, respectively, in P1. After dilution and soot addition this mixture was (of α-pinene SOA, Argentinian soil dust, and graphite soot) 10%, 18%, and 72% in P2. The fractions were estimated based on (a) the total particle number concentration ($C_n$) measured by CPC, (b) an assumption of equal wall loss rate for all the particles, and (c) an assumption of no formation of new particle types.

## 3 Results and discussions

### 3.1 SPMS performance

### 3.1.1 Particle size and detection efficiency

Polydisperse samples were used during FIN-01. The particle size measured by the SPMSs was $d_{va}$ in the free molecular regime ($d_a$ measured by PALMS is assumed equivalent to $d_{va}$) while the ancillary instruments measured the electrical mobility diameter ($d_m$, SMPS) in the transition regime or aerodynamic diameter in continuum regime ($d_{ca}$, APS). The differences

between $d_{va}$, $d_{ca}$, and $d_m$ are due to non-unity density and non-spherical shape factors of the samples. A comprehensive discussion of the differences can be found e.g., in DeCarlo et al. (2004) and Slowik et al. (2004). To facilitate comparison, we converted $d_m$ and $d_{ca}$ to $d_{va}$ using the following equations. It is assumed particles have no voids and slip correction is ignored:

$$d_{va} = d_m \times f_m = d_m \times \frac{\rho_p}{\rho_0 \chi_t \chi_v} \qquad (1)$$

$$d_{va} = d_{ca} \times f_{ca} = d_{ca} \times \sqrt{\frac{\rho_p}{\rho_0}} \times \frac{\sqrt{\chi_c}}{\chi_v} \qquad (2)$$

where $\rho_0$ is unit density (1 g cm$^{-3}$), $\rho_p$ is particle density and $\chi_c$, $\chi_t$, and $\chi_v$ are the dynamic shape factors (DSFs) in the continuum, transition, and free molecular regime, respectively. The values of the parameters used in this study are listed in

Table 2. $f_m$ and $f_{ca}$ are defined as the conversion factors for $d_m$ and $d_{ca}$, respectively. Additional detail can be found in DeCarlo et al. (2004).



Figure 3 shows the normalized particle size distributions measured by the SPMSs for the different particle types. The SPMSs all measured a similar size range, spanning parts of the accumulation (~0.1 to 1 μm) and coarse (> 1 μm) modes. The similar lower and upper cut-off sizes of ~100–200 nm and ~1–3 μm, respectively, were due to similar focusing inlets and similar

wavelengths of detection lasers. The exception is LAAPTOF, which measured particles in a size range of ~450 nm to 2 μm $d_{va}$ during FIN-01, most likely due to non-optimal detection laser alignment. Detection reduction at small and large particle sizes is due to light scattering and focusing of the particle beam, respectively (Bohren and Huffman, 1998; Schreiner et al., 1999).

Overall, the particle size distributions of the minerals (K-feldspar and illite NX) and dusts (desert and soil) measured by the

SPMSs agreed reasonably well with those measured by the SMPS. The size measurements by the SPMSs and SMPS were not always as consistent as for the minerals and dusts. For Snomax this was most likely because of the broader size range, with half of the particles smaller than 100 nm $d_{va}$ (i.e., beyond the typical lower size detection limit of SPMSs). Disagreement may also be attributed to the simplifications inherent in equations (1) and (2) or assumptions about the dynamic shape factors and/or particle density.

Another exception was the experiment with propane soot, for which most of the particles were below the detection limit of most of the SPMSs. The conversion of SMPS data from $d_m$ to $d_{va}$ using equations (1) and (2) for propane soot was more uncertain than for the other particle types. Note that it is challenging to perform size conversions for aspherical particles; even for monodisperse (mass- or mobility-selected) particles, the $d_{va}$ size distributions are broad and asymmetric due to the presence of particles with different shapes and/or orientation-dependent DSF in the free molecular regime (Beranek et al., 2012;

Zelenyuk et al., 2008a). The propane soot particles in the experiments with high O:C ratio were compact and non-spherical, but not fractal, for which a conversion from $d_m$ to $d_{va}$ would be possible as discussed in previous studies (Naumann, 2003; Shapiro et al., 2012; Suski et al., 2021). Moreover, the size distribution of the soot particles evolved during the measurement due to aging/coagulation/compaction (Bhandari et al., 2019; Corbin et al., 2023). Size conversion of soot is beyond the focus of this study. A comprehensive discussion of physical and morphological parameters of soot particles can be found in the

literature (Schneider et al., 2006; Shapiro et al., 2012; Sorensen, 2011; Suski et al., 2021).

Figure 4 shows the DE, as a function of aerosol particle size and separated for each particle type, for the SPMSs. Given the different methodology applied for detection and mass spectral acquisition by miniSPLAT, we compare the results from the other SPMSs first. DE, a strong function of particle size, spans 2–3 orders of magnitude. PALMS exhibited the highest DE for all particle types, especially at the larger sizes (> 400 nm $d_{va}$). At sub-micrometer diameters, ALABAMA and ATOFMS

exhibited comparable DE. This aligns with the respective size ranges corresponding to $DE_{max}$ for PSL for each instrument (Table 1). As mentioned previously, LAAPTOF tended to measure relatively larger particles (> 500 nm) more effectively. Note that particle type also played a role in particle detection, most likely related to a composition dependent shape factor and/or light scattering efficiency. DE for propane soot particles is not shown in Fig. 4 due to the aforementioned low $C_n$ within the detectable size range of the SPMSs and the difficulty of size conversion. MiniSPLAT exhibited relatively higher DE than

the other SPMSs. As previously mentioned, miniSPLAT has defined DE and HR differently in the literature than for the other





SPMSs. Using the same definitions as the other SPMSs, for the corresponding measurements by miniSPLAT in Fig. 4, DE was ~38% of the whole size distribution while HR was ~11%. As an example in Fig 4a), both whole size distribution and mass spectra related DEs are shown to demonstrate the upper and lower limits of miniSPLAT DE.

### 3.1.2 Mass spectra

Examples of average mass spectra, normalized to maximum ion signal (peak area), for the polydisperse aerosol samples are shown in Fig. 5 (Snomax) and Fig. 6 (Moroccan desert dust). These two aerosol types were chosen to illustrate the performance of the different instruments. Snomax is a chemically homogeneous aerosol (Kanji et al., 2017; Möhler et al., 2007; Murray and Liu, 2022). Thus, potential differences between the individual SPMSs due to composition-dependent ionization efficiency should be minimal. Desert dust particles, such as the Moroccan sample, are chemically more complex and diverse (Hoose and

Möhler, 2012; Kandler et al., 2007; Kanji et al., 2017; Marsden et al., 2019; Murray et al., 2012; Murray and Liu, 2022). Average mass spectra for the other aerosol types are shown in Figs. S1–5. In general, the SPMSs had common markers for specific particle types which are summarized in Table 3.

To better compare the mass spectra and quantify commonalities and differences, we conducted a statistical analysis using Pearson's correlation coefficient, denoted hereafter as 'r'. The coefficient measures the strength and direction of a linear

relationship between two variables or data sets, e.g., two average mass spectra. A value of r=1 and -1 indicate a perfect positive and negative linear relationship, respectively, whereas r=0 indicates no linear correlation. The average r for the average positive and negative mass spectra of the different aerosol types between the different instruments can be found in Table S1. The five instruments exhibited comparable markers and good correlation, defined here as a r > 0.6 (greater than 60% linear correlation). On average and across all samples, the r value for positive spectra $r_{avg-pos} = 0.74 \pm 0.12$ and for negative spectra $r_{avg-neg} = 0.67$

$\pm 0.22$.

Figure 7 shows a detailed analysis where we compare the linear correlation coefficient of each SPMS with each other, differentiated by aerosol type. Among the sampled particle types, the strongest correlation was for Snomax, $r_{avg-pos} = 0.92 \pm 0.04$ and $r_{avg-neg} = 0.90 \pm 0.05$. This was a consequence of the aforementioned factors: 1) Snomax was chemically homogeneous, 2) all instruments exhibited common spectral markers with a similar pattern (Fig. 5) and 3) Snomax produced multiple ion

markers, many of which were of high signal/intensity.

The second strongest correlation, in negative spectra, was found for the Moroccan desert dust (Fig. 7 and Table S1). The strong correlation was due to a strong silicate pattern (m/z 60 $SiO_2^-$, 76 $SiO_3^-$, and 77 $HSiO_3^-$; Fig. 6). This was generally true for the desert (Fig. 6) and soil dusts (Fig. S1) which had many common markers in their mass spectra. Despite similar cation markers, the correlations for the positive spectra of the dusts were not as good as negative spectra (Fig. 7 c and d). The major peaks,

e.g., 27 $Al^+$, and/or 39 $K^+$, and/or 56 $Fe^+$ in soil dust had a higher average intensity than markers in desert dust. It is worth noting that soil dusts exhibited more diversity of signal, consistent with their being internal mixtures of minerals and organics, hence a more complex composition (Kögel-Knabner et al., 2008; O'Sullivan et al., 2014; Tobo et al., 2014). As an example, negative spectral organic acids makers, e.g., 45 $COOH^-$, were found in soil dust but not in desert dust.



For the pure mineral samples, e.g., K-feldspar (Fig. S2) and illite NX (Fig. S3), all instruments had common markers at m/z
23 $Na^+$, 27 $Al^+$, 39/41 $K^+$, 56 $Fe^+$, 60 $SiO_2^-$, 76 $SiO_3^-$, and 77 $HSiO_3^-$. Illite NX spectra also contained m/z 24 $Mg^+$. Nitrate and
sulfate markers were found in the negative spectra measured by ALABAMA, especially for K-feldspar. They were identified
as instrument-specific contamination, from an unknown source. This resulted in poor linear correlations with the other
instruments.

For α-pinene SOA, the instruments showed strong correlations in their positive and negative spectra (Fig. 7 and Table S1).
Common features included m/z 12 $C^+$, 24 $C_2^+$ and 36 $C_3^+$, 39 $C_3H_3^+$, 41 $C_3H_5^+$, and 43 $C_3H_7^+$/$C_2H_3O^+$ (Fig. S4). For α-pinene
SOA-coated K-feldspar, the spectra were similar to (uncoated) K-feldspar but with additional organics markers (Fig. S5). It is
worth noting that the correlations for SOA-coated K-feldspar were better than SOA, especially in negative spectra ($r_{avg-neg}$ =
$0.73 \pm 0.03$ and $0.30 \pm 0.29$ for SOA coated K-felspar and SOA, respectively). This was likely due to minerals having relatively
stronger spectral patterns than organics.

Relatively poor correlations were found for propane soot ($r_{avg-pos}$ = $0.51 \pm 0.23$ and $r_{avg-neg}$ = $0.35 \pm 0.26$, average spectra in Fig.
S6). This can be attributed to the small particle size and resulting low data quantity, e.g., only 30 and 73 spectra were collected
by the ATOFMS and by the PALMS instrument, respectively. For the other particle types, thousands of spectra were typically
used for averaging. Given multiple common spectral markers, i.e., pure carbon ions, $C_m^+$ and $C_m^-$ (the number of carbon atoms,
m, can reach >7), better correlation would be expected for soot particles with $d_{va}$ > 200 nm.

One conclusion of this intercomparison is that, in general, spectra were comparable across instrument types. Spectra of particle
types with compounds that created distinct marker peaks compared better than particles of higher compositional diversity
which created spectra with less distinct patterns. There were cases where spectral patterns were instrument-specific, which
resulted in lower correlations, but in at least one case this appeared related to an instrument-specific contamination issue.
Correlation was largely independent of ionization laser wavelength, but some differences were apparent; for example, mineral
and desert dust samples in miniSPLAT and dust samples in ATOFMS did not produce similar positive spectra for the same
particle types (Figs. 5, S2, and S3, and the correlation results for positive spectra in Fig. 7 a, b, and c).

Particle type cross-correlations are shown in Fig. 8 to demonstrate each SPMS's ability to distinguish particle types. Good
correlations (r > 0.8 for most cases) were observed between similar particle types, namely mineral samples (K-feldspar and
illite NX) and dust samples (Moroccan desert dust and Argentinian soil dust). SOA was clearly separated, as was propane soot.
Snomax was not clearly separated from minerals (such as K-feldspar and illite NX) and dusts in the positive spectra due to
common marker peaks (e.g., m/z 23 $Na^+$ and 39 $K^+$), whereas it was distinguishable in negative spectra due to silicate
signatures, which were present for minerals and dusts, but not for Snomax. This highlights the importance of simultaneous
acquisition of spectra of both polarities (a significant limitation of monopolar instruments such as PALMS).

Another important aspect of mass spectra is dynamic range of ion signals and signal-to-noise ratio (SNR). A wider dynamic
range allows detecting and distinguishing ion signals of varying intensities. This enables identification of trace species,
providing useful information in tracking the source of particles (Murphy, 2007). In general, the SPMSs exhibited wide dynamic
ranges and high SNRs in their mass spectra, but variations existed among different instruments. Taking Snomax as an example



(Fig. 5), the dynamic ranges of the SPMSs were ~$10^3$ and ~$10^2$ in positive and negative spectra, respectively. SNR values were similar (~$10^4$) in both positive and negative spectra of PALMS, miniSPLAT and ATOFMS. Relatively lower SNRs (~$10^3$) were found for ALABAMA and LAATOF, with their positive spectra exhibiting higher SNR than negative ones.

## 3.2 Blind experiment comparisons

### 3.2.1 Particle detection and sizing

The two distinct measurement periods, P1 and P2, for the blind experiments were described previously. Unlike the propane soot used in the other experiments, the particles used in blind experiments were pure EC graphitic soot (Crawford et al., 2011). The total $C_n$ during P1 was ~2000 to 3000 cm$^{-3}$ and decreased to ~600 to 900 cm$^{-3}$ during P2, after the dilution and soot addition described previously (Fig. 2).

As shown in Fig. 9 a1) and a2), the size distributions measured by the ALABAMA and the ATOFMS were most comparable. The smaller (sub-200 nm) and larger (> 500 nm) particles were best resolved by miniSPLAT and PALMS. PALMS resolved $d_{va}$ >1.2 µm (based on a comparison with APS data; not shown). LAAPTOF utilized a "first laser mode" (i.e., the ionization laser was triggered immediately after particle detection by the first detection laser) during the blind experiments to obtain more spectra, and, as a result, did not record size information. The other SPMS measurements all agreed in the $d_{va}$ range of ~200 to 800 nm. (Fig. 9 a1 and a2) show an increase in DE for most measurements as particle sizes increased (Fig. 9 b1 and b2). With the exception of miniSPLAT, the normalized size distributions measured by the SPMSs in P1 and P2 do not have significant differences (Fig. 9 a1 and a2). All the DEs were higher in P2 (Fig. 9 b2) than P1 (Fig. 9 b1). The $d_{va}$ size distributions measured by miniSPLAT extended to smaller sizes than the other instruments and exhibited a mode at ~35 nm (see the insert in Fig. 9 a1 and a2), which corresponded to fractal soot particles. As previously mentioned, the $d_{va}$ of fractal soot particles is nearly independent of mass and $d_m$ and is instead determined by the size of primary spherules that comprise fractal agglomerates. The fractal graphitic soot particles used in the blind test were shown to be comprised of primary spherules with diameter of 6.6 ± 1.7 nm (Wenzel et al., 2003). The $d_{va}$ of the graphitic soot particles (~35 nm) is smaller than the $d_m$ of these particles, which varied between 150 nm and 400 nm.

### 3.2.2 Identification of different particle types

Based on independent measurements and data analysis, the participants of the blind experiments identified different particle types in the unknown aerosol mixture. Participants were only told that particles in the blind experiments had been used in the prior FIN-01 experiments but not the number of types, size distributions, or number densities. The number fractions of the identified particle types reported are shown in Fig. 10. Five particle types, including α-pinene SOA (C1), α-pinene SOA-coated minerals (C2), minerals (K-feldspar and/or illite NX) (C3), dust (soil or mineral) (C4), and soot (C5), were identified by the participants. Across both periods, all distinct particle types of the blind experiment, five were reported by PALMS, four by ALABAMA (C1, 3, 4 soil and 5) and ATOFMS (C1, 2, 3, and 5), three by miniSPLAT (C1, 4 soil, and 5) and two by





LAAPTOF (C1 and 4 mineral). SOA fractions measured by PALMS, ALABAMA, and LAAPTOF showed general good
agreement.

### 3.2.3 Discussion of blind experiment results

Size distribution and the DE results in the blind experiments were consistent with the size and size-dependent DE discussion in Section 3.1.1. Some disagreement between SMPS and SPMSs could result from the simplifications inherent in equations (1) and (2), and/or inaccurate assumptions about the dynamic shape factors and/or bulk density. The main difference between
P1 and P2 was the addition of fresh soot particles (Fig. 2).

Of the different particle types present in the APC chamber during the blind experiments, all instruments identified α-pinene SOA. All instruments, with the exception of LAAPTOF, also identified soot. This is most likely due to the aforementioned LAAPTOF low HR at small sizes. The case of soil dust was more complicated. PALMS, ALABAMA, and miniSPLAT all identified different types of soil dust. LAAPTOF identified mineral dust but did not specify the type. ATOFMS identified a
mixture of minerals (illite NX and K-feldspar). ALABAMA and PALMS also identified illite NX and K-feldspar, respectively. This highlights the ability of the SPMSs to generally identify minerals and dusts but the inability to accurately distinguish specifically between dust and mineral types (see discussion in Section 3.1.2). PALMS and ATOFMS both identified SOA-coated minerals, although this type was not added to the chamber. Such internally mixed particles most likely resulted from coagulation of the external mixture of α-pinene SOA with soil dust in the chamber. The identification of such internally mixed
particles is important because it demonstrates SPMSs' capability to investigate processes such as particle aging on a particle-by-particle level. Note that Fig. 10 shows two miniSPLAT-derived pie charts. The left pie chart corresponds to the same data presented by the other SPMSs, the total mass spectra acquired and assigned to various particle classes. The right pie chart uses composition and morphology-dependent detection probabilities, as determined by miniSPLAT for multiple particle types. This calibration yields the 2$^{nd}$ pie chart which can be compared to the inserts in Fig. 2. Note that the other instrument groups did
not produce a similar calibrated pie chart.

### 4 Conclusions and outlook

We present here an intercomparison of five different SPMSs during the FIN-01workshop at the AIDA facility in November 2014. Due to the common fundamental components of the instruments, there was a general agreement in sizing of samples and the mass spectra they produced.
Similar size ranges, typically from ~100–200 nm to ~2–3 μm $d_{va}$, were measured. Overall, DE was found to be instrument-specific and more dependent on particle size than particle type. This highlights the importance of characterizing the size dependence of DE.

The SPMSs exhibited a wide dynamic range (up to ~$10^3$) and high SNR (up to ~$10^4$) in mass spectra. Good linear correlations of spectra measured by different instruments were found ($r_{avg-pos}$ = 0.74 ± 0.12 and $r_{avg-neg}$ = 0.67 ± 0.22), with the best





390 correlations for Snomax ($r_{avg-pos}$ = 0.92 ± 0.04 and $r_{avg-neg}$ = 0.90 ± 0.05). The lowest correlation was found for propane soot ($r_{avg-pos}$ = 0.51 ± 0.23 and $r_{avg-neg}$ = 0.35 ± 0.26) due to the low data quantity resulting from the size below the typical SPMS detection limit. Particle identification favored the bipolar instruments.

Instrument-specific ability to differentiate particle type was evaluated using both cross-correlations and validation in blind experiments. It is shown that all SPMSs were able to differentiate between SOA, soot, and soil dust, but had difficulties

395 distinguishing between specific dusts/minerals. The results of the blind experiments show that SPMSs can detect changes in particle mixing state, which in our experiments likely resulted from coagulation of externally mixed particles. The results should help our community advance our understanding of such instruments and their potential for investigating atmospheric particles and processing.

In recent years, SPMSs have been continuously upgraded in both hardware (particle sampling, optical design, and mass

400 spectrometer) and software (data acquisition and data analysis methods). Since the FIN-01 workshop, participants and other groups have been improving hardware as well as data analysis processes and particle type retrievals. For example, a method for distinguishing K-feldspar from illite has been developed by Marsden et al. (2018) and subsequently used in analyzing ambient data (Marsden et al., 2019). Since not all the minerals have the same properties, e.g. ice nucleating ability (Atkinson et al., 2013), the capability to distinguish minerals is critical for the research field. Future investigations could consider

405 improving our ability to further distinguish particles, minerals, and dusts specifically.

Particle mass quantification, unaddressed in this paper, is another important contemporary topic. Some achievements have been made in the SPMS community, but further studies are still in high demand. Therefore, a new workshop could focus on quantification. As new SPMSs, new software and data analysis methods are developed, we propose that intercomparison workshops in laboratory should continue, and ideally, should also include field activities due to complexity of ambient

410 particles.

***Data availability.*** All data used for the figures in this paper can be accessed at

415 https://radar.kit.edu/radar/en/dataset/ZGLkxMLhgxxEHhkk?token=rgHxTfHgvyPJeZZAdQQq. The DOI of the dataset is: https://doi.org/10.35097/ZGLkxMLhgxxEHhkk and will be available upon publication (Shen et al., 2024).

***Author contributions.*** XS analyzed data, produced tables and figures, and was the lead manuscript writer. DJC participated in FIN-01, provided suggestions for the data interpretation and discussion, reviewed, and edited the manuscript. DMB, FM,

420 NAM, JS, JW, MAZ, and AZ operated the SPMSs and did basic analysis of the data. HS contributed to the generation and coating of reference particles, particle size data and discussion. NH helped to coordinate the campaign. CM, DMM, and OM contributed as the referees in the blind experiments and provided scientific support for evaluating, interpretation and discussion



of the results. DJC, OM, and PJD organized the FIN-01 workshop. HC gave general comments on this paper. All authors contributed to the final text.


*Competing interests.* The authors declare that at least one of the (co-)authors is a member of the editorial board of Atmospheric Chemistry and Physics. The authors also have no other competing interests to declare.

*Special issue statements.* This article is part of the special issue "Fifth International Workshop on Ice Nucleation (FIN)".


*Acknowledgements.* The FIN-01 workshop was supported by the U.S. National Science Foundation grant no. AGS-1339264, and by the U.S. Department of Energy's Atmospheric System Research, an Office of Science, Office of Biological and Environmental Research program, under grant no. DE-SC0014487, and by the German Science Foundation DFG (FOR 1525 "INUIT"). Special thanks to the AIDA staff at KIT for operating the chambers and associated instrumentation, and general

technical. We thank Susan Schmidt's work on operating the ALABMA instrument and analyzing the data, Andreas Hünig for support during the measurements. We thank Berko Sierau and Ulrike Lohmann for their support with ATOFMS data analysis and employment. We also thank participants from other SPMS teams that participated in FIN-01 but did not provide data for this manuscript: the ATOFMS team at the University of California, San Diego (UCSD), LAAPTOF team at Vienna University, the LAAPTOF team at Carnegie Mellon University (CMU), and the SPLAT team at Max Planck Institute for Chemistry

(MPIC).

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





**Table 1: SPMSs, performance parameters, and data analysis methods**

| | PALMS | ALABAMA | miniSPLAT | LAAPTOF (AeroMegt, Germany) | ATOFMS (TSI 3800-100, USA) |
|---|---|---|---|---|---|
| **Affiliation** | MIT | MPIC | PNNL | UoM | ETH |
| **Inlet Flowrate (cm³ min⁻¹)** | ADL 400 | ADL 80 | ADL 100 | ADL 100 | ADL 100 |
| **Detection and sizing** | One Nd:YAG laser Two beams (532 nm, 50 mW, beam width: 1st 200 µm, 2nd 100 µm) | One laser diode Two beams (405 nm, 50 mW, beam width: 190 µm) | Two Nd:YAG lasers (532 nm, 300 mW, beam width: 330 µm) | Two laser diodes (405 nm, 40 mW, beam width:> 50 µm) | Two Nd:YAG laser (405 nm, 95 mW, beam width: ~ 0.5 mm) |
| **LDI** | One step ArF Excimer laser (193 nm, 3–5 mJ/pulse, pulse duration: 4 ns, beam width: 150 µm at focal spot; beam direction: perpendicular to particle beam) | One step Nd:YAG laser (266 nm, 3-4 mJ/pulse, pulse duration: 5.2 ns, beam width: 250 µm, beam direction: facing particle beam) | One step ArF Excimer laser (193 nm, 0.5–2 mJ/pulse, pulse duration: 8 ns, beam focal spot: 550 × 750 µm, laser beam direction: sideways to the particle beam) | One step ArF Excimer laser (193 nm, 4 mJ/pulse, pulse duration: 6–8 ns, beam width: 300 µm, beam direction: facing particle beam) | One step Nd:YAG laser (266 nm, max 5 mJ/pulse, but 1.1–1.2 mJ/pulse in FIN-01, pulse duration: 5 ns, beam width: 2.4 mm; beam direction: perpendicular to particle beam) |
| **Distances** | Inlet exit – 1st laser's focal point: 55 mm | 90.5 mm | N/A | N/A | N/A |
| | 1st – 2nd lasers: 34 mm | 70 mm | 109 mm | 113 mm | 6 cm |
| | 2nd – 3rd lasers: 100 µm | N/A | N/A | N/A | 12 cm |
| **MS** | Unipolar TOF (Switchable polarity) | Bipolar TOF | Bipolar TOF | Bipolar TOF | Bipolar TOF |
| **Size range** | 200 nm – 3 µm $d_{va}$ | 200 nm – 1 µm $d_{va}$ | 50 nm – 1.4 µm $d_m$ | 200 nm – 3 µm $d_{va}$ | 100 nm – 3 µm $d_{va}$ |
| **DE for PSL** | min: 0.1 % (150 nm) max: 10% (300 nm – 1 µm $d_{va}$) | 0.3% (200 nm) 86% (400 nm) | ~0.02% (50 nm $d_m$) ~100% (125 – 600 nm $d_m$) | max=1% at 600 nm (UoM LAAPTOF) | 0.5% (90 nm) 47.4 % (290 nm) |
| **HR for PSL** | ~95% (no obvious size dependent and not limited to spherical particles, e.g., PSL) | min: ~1% (200 nm) max: ~75% (400 nm) | ~100% (125 – 600 nm) | max: ~70% (UoM LAAPTOF) | ~57% (90 nm) ~93% (290 nm) |
| **Mass resolution** | ~200 (max) | Between 100 and 600, ~400 for m/z 200 | N/A | >600 at m/z 184 | ~500 for m/z 1 to 800 |
| **Classification method** | Hierarchical cluster algorithm & Target-oriented particle typing | Fuzzy c means clustering | K-means clustering | Fuzzy c-means clustering | ART-2a algorithm Other classification methods available as part of e.g., FATES |
| **References[a]** | Cziczo et al., 2006 | Brands et al., 2011 Schmidt et al., 2017 | [b]Zelenyuk et al., 2015 | Gemayel et al., 2016 Marsden et al., 2016 Shen et al., 2018 | Prather et al., 1994 Gard et al., 1997 Su et al., 2004 |


[a]Note that the references for the commercial instruments are not limited to the specific ones used in this study. [b]Note that the terminology in DE and HR in the miniSPLAT reference is different from the other SPMSs (see text for details).



**Table 2: Aerosol types, properties, composition, and generation techniques**

| Aerosol type | Density[a] (g cm$^{-3}$) | DSF[a] ($\chi_c$; $\chi_t$; $\chi_v$) | Composition | Generation method | AIDA Exp # | ACP Exp # |
|---|---|---|---|---|---|---|
| K-feldspar (FS01) | 2.56 | 1.2; 1.2; 1.5 | FS01: microcline 76%, albite 24% | Rotating brush (PALAS, RGB1000) | | 6 |
| Illite NX | 2.65 | 1.4; 1.4; 1.9 | illite 69%, kaolinite 10%, calcite 3%, quartz 3%, feldspar (orthoclase/sanidine) 14% | Rotating brush (PALAS, RGB1000) | | 13, 21 |
| Desert dust[b] (Moroccan) | 2.50 | 1.5; 1.5; 1.8 | Quartz 50-60%, illite (<10% total feldspar), iron oxide, calcite, and doromite | Rotating brush (PALAS, RGB1000) | 20 | 26 |
| Soil dust (Argentinian) | 2.60 | 1.4; 1.4; 2.0 | Mixture of minerals and organics | Rotating brush (PALAS, RGB1000) | 48 | 25, 29 |
| Propane soot | 1.40 | 1.8; 1.8; 2.8 | Elemental carbon (EC) and Organic carbon (OC) | Propane burner (RSG miniCAST; Jing Ltd) Incomplete combustion of propane, C/O=0.54 | | 14 |
| Graphite soot | 2.26 | 2.8; 2.8; 5.0 | Pure EC | Graphite spark generator from Palas (GSG 1000) | | 29 |
| α-pinene SOA[c] | 1.25 | 1.0; 1.0; 1.0 | Complex mixture of mainly organic acids and aldehydes | In situ formation from ozonolysis of a-pinene (nucleation and condensation growth) | | 27, 29 |
| SOA-coated K-feldspar (FS04) | N/A | N/A | α-pinene SOA and FS04 FS04: microcline 80%, albite 18%, quartz 2% | In situ formation from ozonolysis of α-pinene and condensation to dust particles | 46 | |
| Snomax[c] | 1.35 | 1.0; 1.0; 1.0 | protein complexes from nonviable *Pseudomonas syringae* bacteria | Atomizer (TSI, 3076) | | 16, 17, 22 |

[a]For most of the particle samples, the particle density and DSF in transitional and free molecular flow regimes, $\chi_t$ and $\chi_v$, were obtained from the measurements by miniSPLAT during FIN-01 as described in Alexander et al. (2016). $\chi_t$ is near the continuum flow limit, for simplicity we assume $\chi_c$ (not measured) is equal to $\chi_t$. [b]For desert dust, particle density and DSF is described in Froyd et al. (2019).  [c]For α-pinene SOA and Snomax sphericity is assumed ($\chi_c = \chi_t = \chi_v = 1$), therefore the densities shown here are the effective density. Additional detail can be found in Zelenyuk (2008a), Saathoff et al. (2009) and Wex et al. (2015), respectively.




**Table 3: Particle types and the corresponding mass spectral markers**

| Particle type | Markers_cations | Markers_anions | Specific patterns | Other potential ions |
|---|---|---|---|---|
| K-feldspar | 23 Na, 27 Al, 39&41 K (28 Si, 44 SiO)* | 60 $SiO_2$, 76 $SiO_3$, 77 $HSiO_3$ (43 AlO, 88 $Si_2O_2$, 103 (AlO)SiO2, 119 $AlSiO_4$, 136 $(SiO_2)_2O$, 148$(SiO_2)_2$Si, 179 $AlSiO_4.SiO_2$) | | 7 Li, 56 Fe, 63 & 65 Cu, 64 & 66 Zn, 85 & 87 Rb, 133 Cs (PALMS), 137 & 138 Ba, 153 &154 BaO |
| Illite NX | 23 Na, 24 Mg, 27 Al, 56 Fe, 39&41 K (28 Si, 44 SiO) | 43 AlO, 60 $SiO_2$, 76 $SiO_3$, 77 $HSiO_3$, 88 $Si_2O_2$, 103 (AlO)SiO2, 119 $AlSiO_4$, 136 $(SiO_2)_2O$, 148$(SiO_2)_2$Si, 179 $AlSiO_4.SiO_2$ 63 $PO_2$, 79 $PO_3$ | | 7 Li, 63 & 65 Cu, 64 & 66 Zn, 85 & 87 Rb, 133 Cs (PALMS), 137 & 138 Ba, 153 &154 BaO, |
| Desert dust (Moroccan) | 7 Li, 23 Na, 24 Mg, 27 Al, 39&41 K, 56 Fe (28 Si, 44 SiO); (40 Ca, 56 CaO, 72 $CaO_2$, 96 $Ca_2O$, 112 $(CaO)_2$); (63 & 65 Cu, 64 & 66 Zn, 85 & 87 Rb, 137 & 138 Ba, 153 &154 BaO) | 26 CN or $C_2H_2$, 42 CNO or $C_2H_2O$, 43 AlO, 60 $SiO_2$, 76 $SiO_3$, 77 $HSiO_3$, 88 $Si_2O_2$, 97 $HSO_4$, 103 (AlO)SiO2, 119 $AlSiO_4$, 136 $(SiO_2)_2O$ 148$(SiO_2)_2$Si, 179 $AlSiO_4.SiO_2$ | | 133 Cs (PALMS) 45 COOH, 71 $CCH_2COOH$ 63 $PO_2$, 79 $PO_3$ |
| Soil dust (Argentinian) | similar as desert dust | In addition to the markers of MD, there are organic acids related ones 45 COOH, 71 $CCH_2COOH$ (89 (CO)OCOOH) | Distinguish desert and soil dusts: soil dust has organic acids, less intensive silicate pattern, and more intensive anion pairs of m/z 26⁻ & 42⁻ and 63⁻ & 79⁻ | |
| Soot with org (Propane soot) | EC fragments: $C_m$ OC fragments: 39 $C_3H_3$, 56 $C_4H_8$ (27 $C_2H_3$, 28 CO, 40 $C_2O$, 41 $C_3H_5$, 44 COO, 50 $C_4H_2$, 69 $C_5H_9$) | EC fragments: $C_m$ OC fragments: 26 $C_2H_2$, 42 $C_2H_2O$ | One fork shape at m/z 12⁺, 24⁺, 36⁺ EC pattern is more intensive than OC fragments | |
| α-pinene SOA | mainly $C_xH_y$ and $C_xH_yO_z$ fragments: e.g., 12 C, 13 CH, 15 $CH_3$, 19 $H_3O$, 24 $C_2$, 27 $C_2H_3$, 28 CO, 36 $C_3$, 39 $C_3H_3$, 41 $C_3H_5$, 43 $C_3H_7$ or $C_2H_3O$, 55 $C_4H_7/C_3H_3O$, 59 $C_2H_2OOH$, 69 $C_5H_9$, 77 $C_6H_5$, 83 $C_6H_{10}/ C_5H_7O/C_4H_2OOH$, 85 $C_7H_5$, 91 $C_7H_7$, 95 $C_7H_{11}$ | mainly organic acids fragments: e.g., $(CH_2)_{n=0-11}$ COOH: 45 to 199 $(CH_2)_{n=0-10}$ CCOOH: 57 to 197 $(CH_2)_{n=0-4}$ (CO)OCOOH: 89 to 145 (Only for PALMS and LAAPTOF) | Two fork shapes m/z 12⁺, 24⁺, 36⁺ & 39⁺, 41⁺, 43⁺ | |
| Snomax | 23 Na, 30 NO, 39 K or $C_3H_3$, 41 K or $C_3H_5$, 47 PO, 56 Fe or $C_4H_8$, 62 $Na_2O$, 70 $C_5H_{10}$, 72 FeO, 78 $Na_2O_2$ 165 $Na_3SO_4$, 181 $Na_2SO_4NaO$ or $C_4H_7O_4NO_3$, 197 $NaK_2SO_4$ (18 $NH_4$, 28 CO, 44 $CO_2$, 86 $(C_2H_5)_2NCH_2$,213 $K_3SO_4$) | 26 CN or $C_2H_2$, 42 CNO or $C_2H_2O$ 45 COOH, 59 $CH_3COOH$, 71 $CCH_2COOH$ 63 $PO_2$, 79 $PO_3$, 96 $SO_4$, 97 $HSO_4$, 119 $NaSO_4$ (135 $KSO_4$, 153 $Na_2Cl_3$) | Phosphate fragments at both pos and neg Important pairs m/z 23⁺&39⁺; 18⁺&30⁺ 26⁻&42⁻; 63⁻&79⁻ | |

Note: ions in parenthesis are not observed in all SPMS mass spectra.



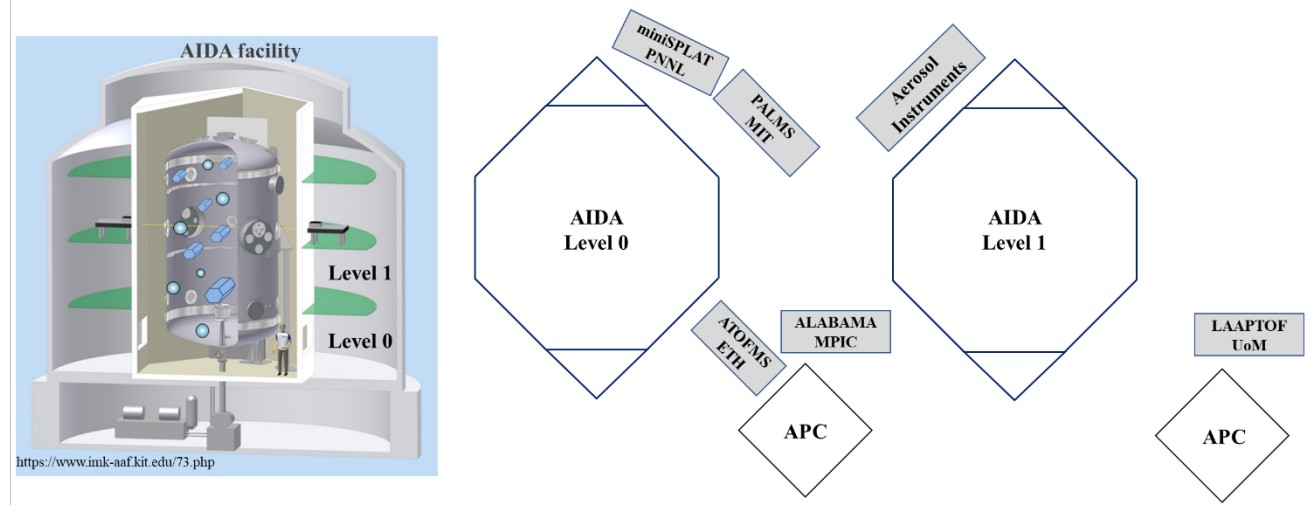


**Figure 1: Experimental setup. During the campaign the instruments were positioned to minimize the length of the sampling lines to the AIDA and APC chambers. The sampling lines were 6 × 4 mm (OD × ID) stainless steel tubing.**

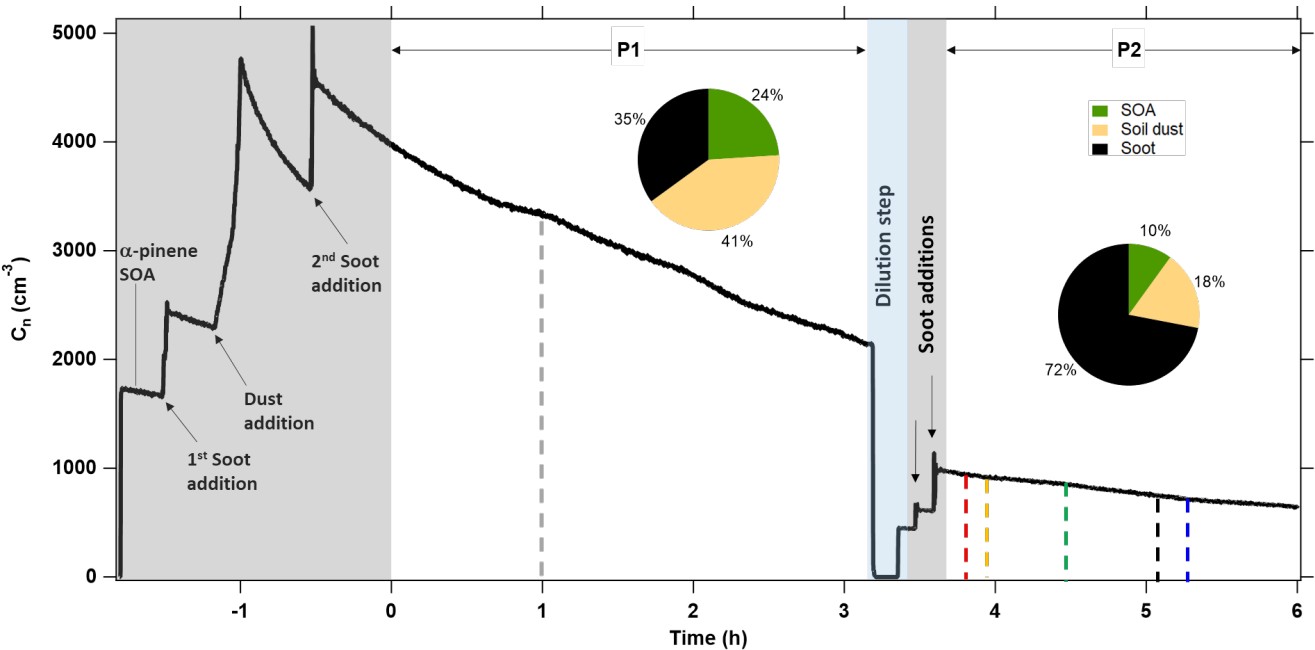


**Figure 2: Time series of total particle $C_n$ measured by a CPC in the APC chamber during the blind experiments. SPMS measurements were conducted during two periods, P1 and P2. In P1, the SPMSs started sampling at the same time (grey dashed line), except miniSPLAT which started 15 min later. In P2, sampling started at different times (red, orange, green, black and blue dashed lines are for PALMS, ATOFMS, LAAPTOF, miniSPLAT and ALABAMA, respectively). The pie charts denote particle**
**types and their number fractions in P1 and P2, respectively. Note that the grey and blue shaded areas denote the aerosol mixture preparation periods, and a dilution step when the chamber was partly pumped out and refilled with clean air.**

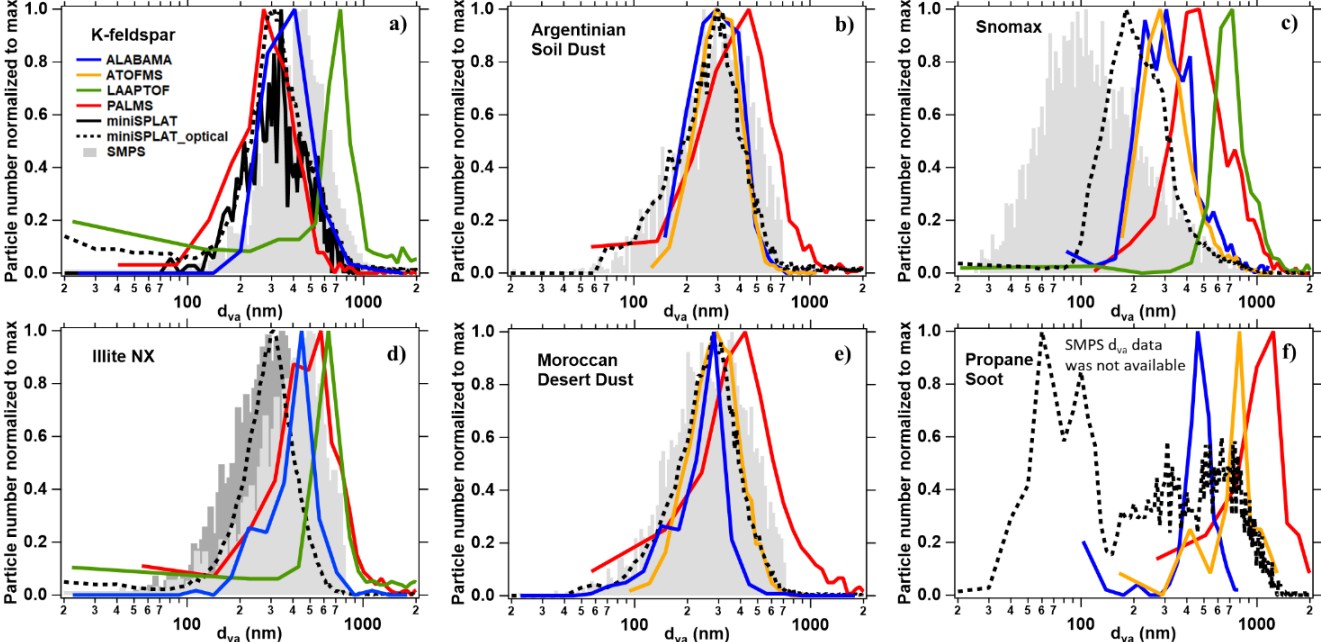

**Figure 3: SPMS size distributions (normalized to maximum particle number) for selected particle types, i.e., a) K-feldspar, b) Argentinian soil dust, c) Snomax, d) Illite NX, e) Moroccan desert dust, and f) Propane soot. Color codes correspond to $d_{va}$ measured**
**by ALABAMA (blue), ATOFMS (orange), LAAPTOF (green), PALMS (red), and miniSPLAT (black). Note that the dashed black curves represent the DE of the size distribution measured by miniSPLAT, while in a) the solid black curve represents DE consistent with the other SPMSs (see text for details). Particle sizes measured by the SMPS over a scan range of 14–820 nm $d_m$ was converted to $d_{va}$ (grey) for comparison. For illite NX particles, two different SMPS size distributions from the APC13 and APC21 experiments are shown in light and dark, respectively (miniSPLAT result was from only APC21, while the others were from APC13). For soot**
**particles, the SMPS $d_{va}$ data was not available for these experiments (see text for details). Dashed lines in d) are used to emphasize the extent of the distributions. In most cases the particle numbers by each SPMS (used to derive the distribution curves) were hundreds to thousands. The exception is propane soot, for which most of SPMSs detected < 100 particles due to the detection limit.**

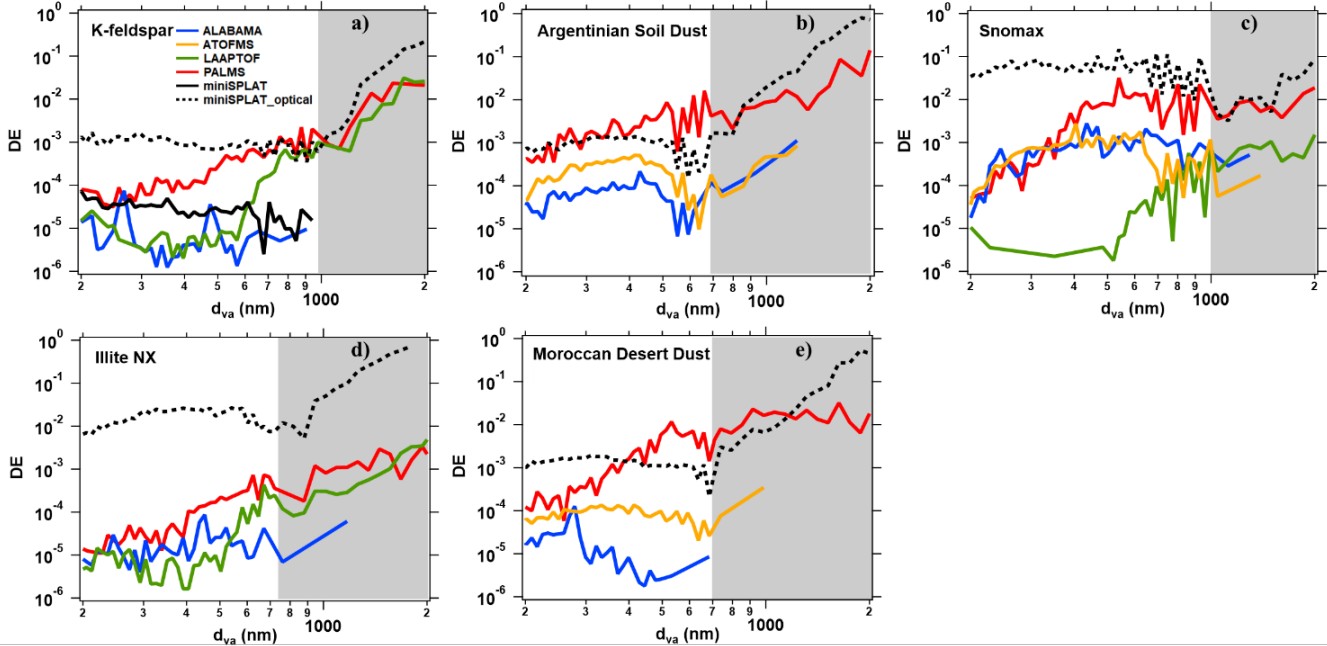

**Figure 4:** Detection efficiencies of SPMSs for selected particle types, i.e., a) K-feldspar, b) Argentinian soil dust, c) Snomax, d) Illite NX, and e) Moroccan desert dust, as a function of the $d_{va}$ measured by ALABAMA (blue), ATOFMS (orange), LAAPTOF (green), PALMS (red), and miniSPLAT (black). Note that the uncertainty for super-micrometer particles is larger due to the low number concentration of these particles in the experiments. The DE curves in white and shaded area are based on SMPS and APS results, respectively. There were overlap results between SMPS and APS for the sizes around 700 nm to 1 μm $d_{va}$. In such an overlap range, we chose SMPS data as reference to calculate DE. Given different shape factors and particle densities, the conversions of $d_m$ and $d_{ca}$ to $d_{va}$ were different for different particle types. Therefore, the end point of SMPS based result or shift point between white and shaded area varies from sample to sample. Noted that the DE of miniSPLAT is defined differently from the other SPMSs. The sized distribution derived DE (dashed black) and the hits only derived DE (solid black) for K-feldspar is shown to provide the upper and lower limits of miniSPLAT DE (see text for details).





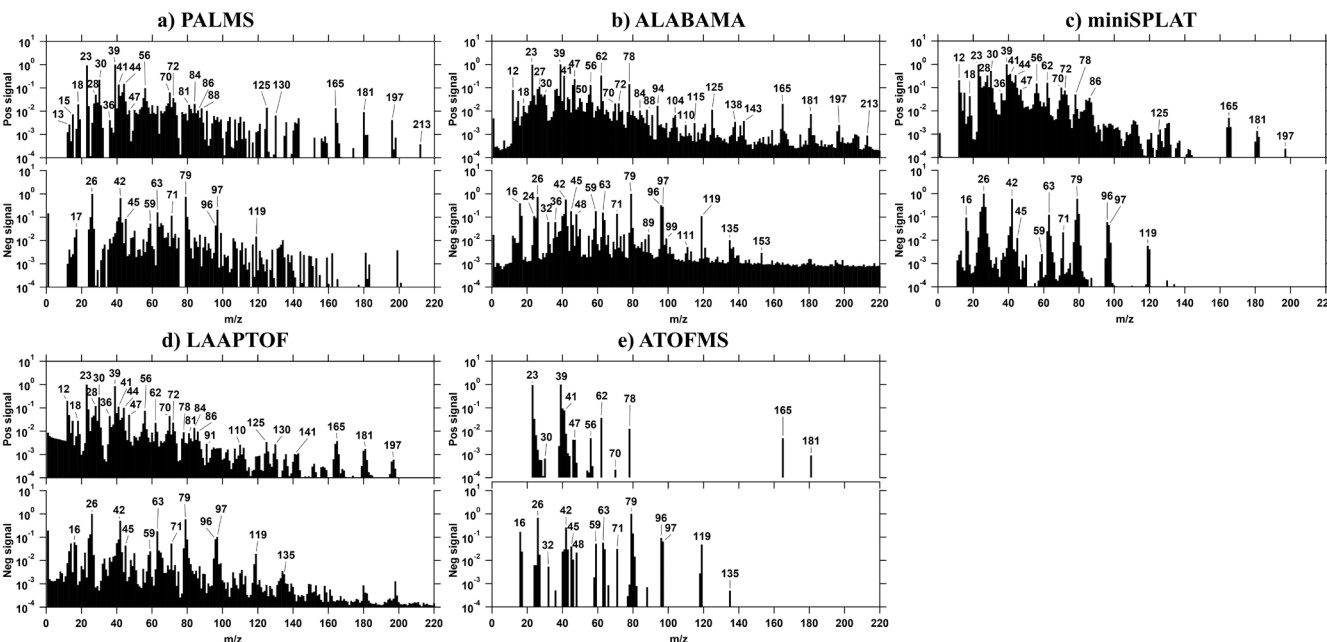


**Figure 5: Normalized average mass spectra of Snomax, measured by a) PALMS, b) ALABAMA, c) miniSPLAT, d) LAAPTOF, and e) ATOFMS. The numbers of spectra averaged for each are 973 a), 1018 b), 2327 (c), 260 d), and 1071 e).**

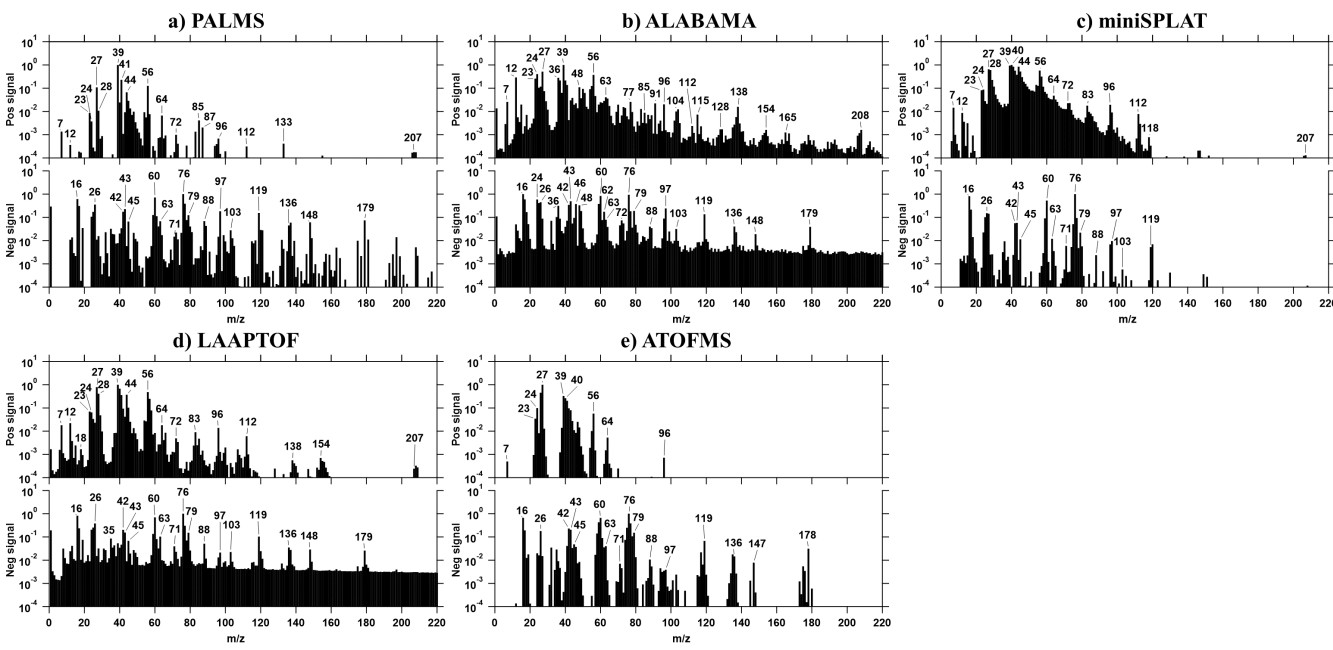


**Figure 6: Normalized average mass spectra of Moroccan desert dust by a) PALMS, b) ALABAMA, c) miniSPLAT, d) LAAPTOF, and e) ATOFMS. The numbers of spectra averaged for each are 715 a), 353 b), 346 (c), 215 d), and 1447 e).**



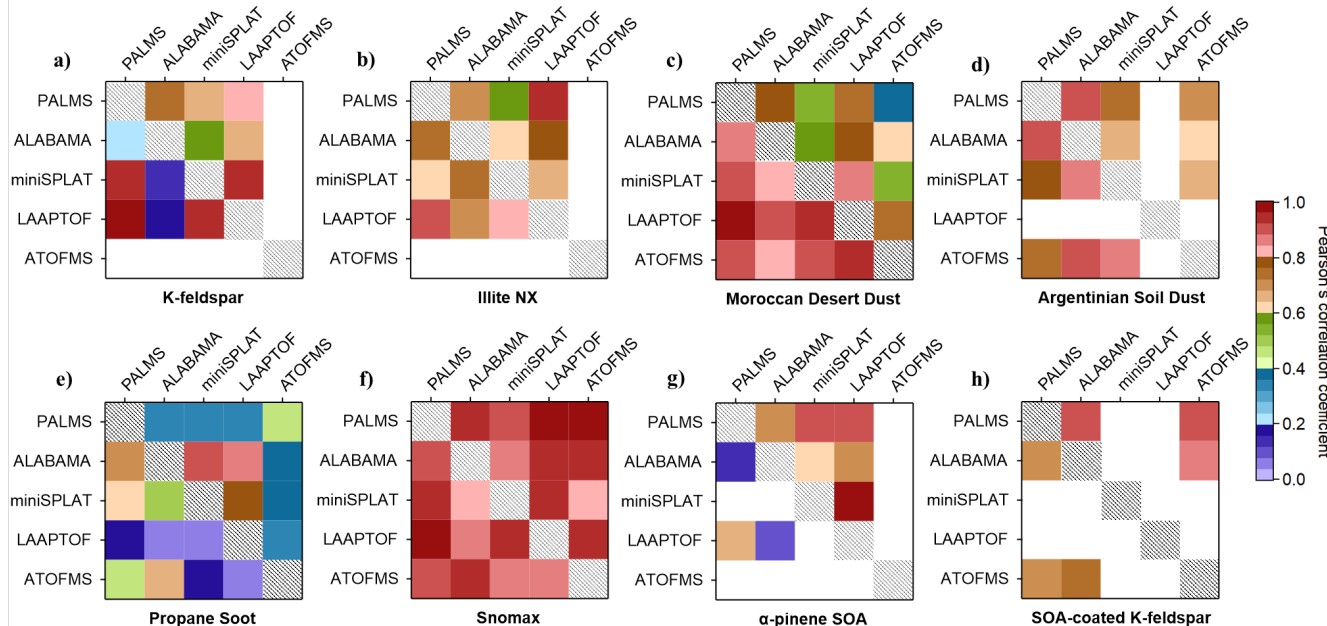

**Figure 7: Correlation plots of average spectra of the different particle samples a) K-feldspar, b) Illite NX, c) Moroccan desert dust, c) Argentinian soil dust, e) propane soot, f) Snomax, g) α-pinene SOA, and h) SOA-coated K-feldspar, analyzed by the five SPMSs.**
**Within each diagram correlation results for positive spectra are upper right and negative spectra are bottom left. White cubes denote cases where no data is available.**



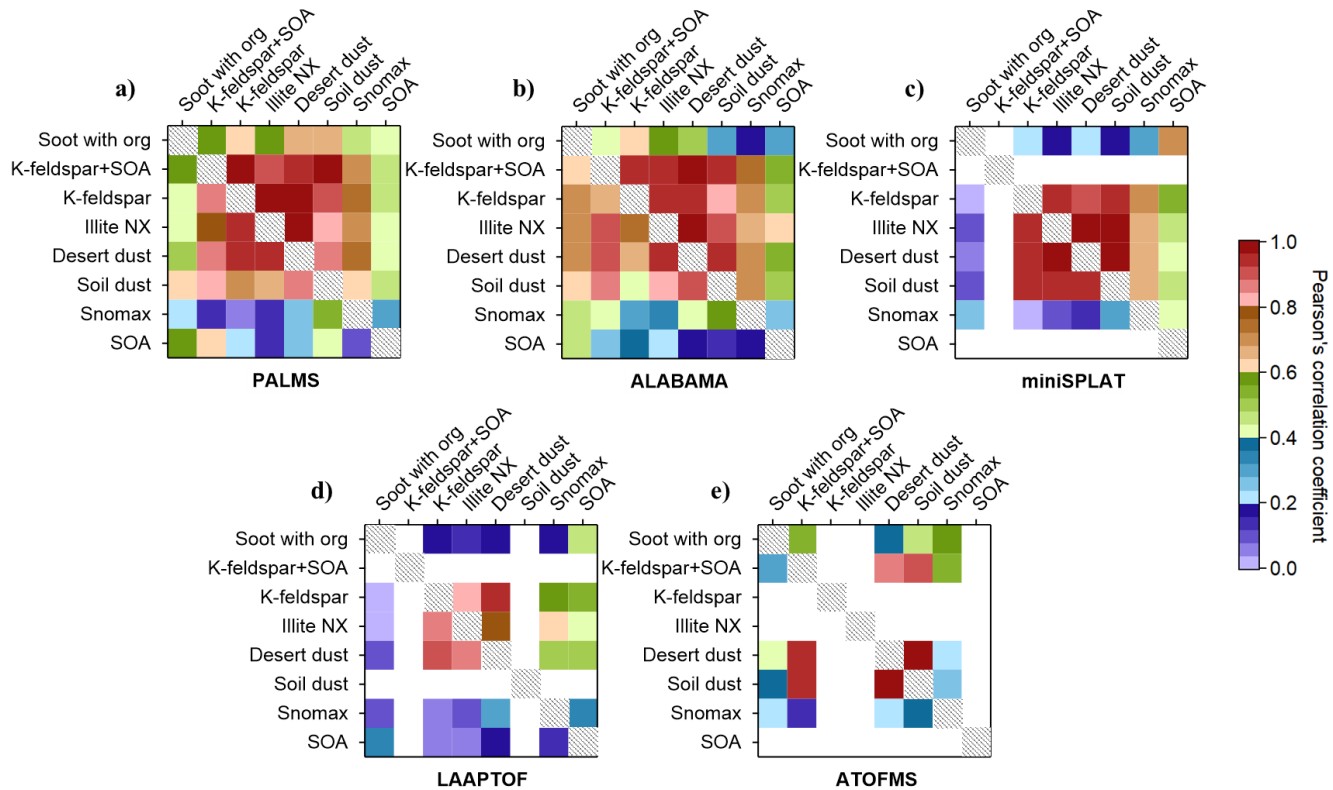

**Figure 8: Correlation plots of average spectra of different samples analyzed by a) PALMS, b) ALABAMA, c) miniSPLAT, d) LAAPTOF, and e) ATOFMS. Correlation results for positive spectra are upper right, while negative spectra are bottom left. Note that, the soot with organics and K-feldspar+SOA denote the propane soot and SOA-coated K-feldspar, respectively.**






**Figure 9: Particle size distribution and DE of PALMS (red), ALABAMA (blue), ATOFMS (orange), and miniSPLAT (black) during two different time periods (P1 and P2) of the blind experiments. For miniSPLAT, the full-size ranges measured during P1 and P2**

**are inserted in a1) and a2), respectively (note that the smallest size mode appears large, due to the lognormal mode in the X-axis). Note that the dashed black curves represent the DE of the whole size distribution measured by miniSPLAT, while in the solid black curve represent DE consistent with the other SPMSs (see text for details). $d_m$ measured by SMPS (grey) was converted to $d_{va}$ for comparison. The DE curves in the white and shaded areas are based on SMPS and APS results, respectively. $f_m$ =1.13 and $f_{ca}$= 1.06, the number weighted values derived from α-SOA and Argentinian soil dust (particle number ratio of 2:3), were used in these**

**experiments. Note most of the soot particles were below the detection limit of SPMSs, with the exception of miniSPLAT (see text for details).**





**Figure 10: Particle classes and their relative contributions for blind periods P1 and P2 for: a1-2) PALMS, b1-2) ALABAMA, c1-2) LAAPTOF, d1-2) ATOFMS and e) miniSPLAT (which only reported data in P1). The plots shown here are the data provided to the referees after experiments (i.e., before participants knew the composition of the blind experiment). Note the left pie chart for miniSPLAT represents acquired mass spectra, consistent with what is presented for the other SPMSs. The right pie chart represents calibrated data (see text for details). Particle clustering is shown for positive and negative spectra separately for the unipolar switchable PALMS instrument.**