# Peer review of "Measurement report: The Fifth International Workshop on Ice Nucleation Phase 1 (FIN-01): Intercomparison of Single Particle Mass Spectrometers"

_EGUsphere, 2024_

## Referee Comment (RC1)

**Review of egusphere-2024-928**

The authors report on a systematic comparison study of single-particle mass spectrometers (SPMS), focusing on particles with high potential for cloud condensation and ice nucleation. This is a large and important task that has been undertaken with great effort. One of the greatest uncertainties of the SPMS technology is the lack of knowledge about the comparability of data sets, which emphasizes the importance of direct comparisons. This unique study is well designed and provides important insights into the performance, practical considerations and a fair and unbiased comparison of the different instruments and concepts. The manuscript is technically sound, interesting and well written; I have no concerns that this manuscript is suitable for publication in ACP.

I found a few minor errors and have some comments that might help to further improve the manuscript:

- From their study, the authors gained detailed insights into some well-known general weaknesses of SPMS technology. In the conclusion/outlook, development directions for improvements and developments beyond the current technology could be discussed. For example, optical detection of all SPMS is limited to particles larger than ~150 nm (and not MS sensitivity) due to light scattering limitations. Strategies to improve quantification were also suggested (and referred to) and could be briefly discussed.
- L43/44: Repetitive wording "We found…"
- The abstract could also be a bit shorter. E.g. one of the studies key finding "We found that instrument-specific DE was more dependent on particle size than particle type." is somehow hidden behind detailed correlation data.
- L89: 405 nm diodes → 405 nm laser diodes
- L99: The two step approaches do not only reduce fragmentation. As part of the ionization occurs in the particle's gaseous plume, quantification can be improved (Woods et al., 2001) and resonance effects can be used to increase the sensitivity to organics (Passig et al., 2022, Schade et al., 2019).
- L240 The LOD refers to the optical detection limit, so consider e.g. "…most of the particles were smaller than the optical detection limit of most SPMSs…."
- L244: was the abbreviation DSF introduced before? Maybe I missed it…
- L253: PALMS has high hit rates because the second detection laser is very close to the ion source. However, this design has also drawbacks, as solid-state ionization lasers cannot be used and the particle detection optic's layout is limited by the ion source. Therefore, ATOFMS and ALABAMA seem to be superior for ultra-fine particles, see e.g. Fig. 3c (Snomax).
- L263: Could the DE values for both modes of the miniSPLAT be shown in table 1?
- L330: Two Typos: "ALABMA" and "LAATOF"
- Table 1: The ATOFMS sizing lasers have probably 532 nm wavelength, if they are Nd:YAG.
- Table 1: Use mm or cm consistently.
- Table 1: Hit Rate PSL for miniSPLAT should also be annotated with footnote "b".
- Table 3: "One fork shape" and "Two fork shape" sound nice but should be put in quotation marks.
- Fig 1: In case you have problems with the length of the manuscript, consider to remove figure 1 as it contains no important information for the reader.

**References**
- Woods, E.; Smith, G. D.; Dessiaterik, Y.; Baer, T.; Miller, R. E. *Anal. Chem.* **2001**, *73*, 2317–2322
- J. Passig, J. Schade , R. Irsig , T. Kröger-Badge , H. Czech , T. Adam , H. Fallgren , J. Moldanova , M. Sklorz , T. Streibel and R. Zimmermann, *Atmos. Chem. Phys.*, 2022, **22** , 1495 —1514
- J. Schade , J. Passig , R. Irsig , S. Ehlert , M. Sklorz , T. Adam , C. Li , Y. Rudich and R. Zimmermann, *Anal. Chem.*, 2019, **91** , 10282 —10288

---

## Referee Comment (RC2)

**Referee comment for egusphere-2024-928 "Measurement report: The Fifth International Workshop on Ice Nucleation Phase 1 (FIN-01): Intercomparison of Single Particle Mass Spectrometers" by Shen et al.**

The manuscript describes the intercomparison and blind testing of five Single Particle Mass Spectrometers (SPMS) that were operated in parallel at the AIDA aerosol chamber. A number of different aerosol types were supplied to the chamber and size distribution measurements as well as mass spectra for the SPMSs were compared. Correlations between the instruments and between the different particle types are discussed. Two blind experiments were conducted where a mixture of three particle types (SOA, soil dust and graphite soot) were supplied to the chamber, but the participants of the blind experiments did not know about the particle types, sizes and relative fractions.

The intercomparison constitutes an important study to evaluate the performance and limitations of SPMS instruments. Although the instruments are demonstrated to be generally capable of identify and classify different aerosol types, the (quantitative) results can vary widely (e.g., Fig. 10, P1, ALABAMA suggests >90% pure SOA particles, while ATOFMS detects Illite NX and K-feldspar in >90% of the particles).

My main comment concerns the discussion of the blind tests. This discussion should be expanded. The authors should add the pie charts of the mixture introduced into the chamber (already shown as inserts in Fig. 2) again in Fig 10. It is noteworthy that the miniSPLAT with calibration is able to accurately reproduce the original particle mix. The discussion of the calibration procedures for miniSPLAT should be expanded and/or a reference for the calibration procedures should be given. Can such a calibration be regularly achieved, e.g., for field measurements? Furthermore, it should be discussed whether such calibration would also be possible for the other instruments. It should be discussed to what extent measurements of the individual SPMS instruments are considered to be not quantitative at all, or semi-quantitative, or quantitative after calibration. Can uncertainties be provided for each instrument? A strong caveat should be added that some seemingly obvious interpretations of such pie charts from SPMS results are not possible, and that such pie-charts can potentially be quite misleading (e.g., ALABAMA classifying >90% of the particles as SOA does not mean that the aerosol is actually dominated by SOA; four instruments seeing less than 5% soot particles does not mean that soot is only a minor component of the aerosol, etc.).

Overall, the manuscript is well written and clear. The manuscript is clearly suitable for ACP after considering my main comment. The comprehensive intercomparison of the performance of the individual SPMS instruments is very insightful and valuable for interpreting SPMS results in general and for understanding the current limitations of each instrument.

Technical correction:

Line 857: "Note that…"

---

## Author Comment (AC1)

Responses to Referees

We gratefully thank the reviewers for their helpful comments to improve the quality of our manuscript. Reviewers' comments are in black. Our point-by-point replies are in blue. Changes to the manuscript text are in green.

**Referee #1 comments:**

The authors report on a systematic comparison study of single-particle mass spectrometers (SPMS), focusing on particles with high potential for cloud condensation and ice nucleation. This is a large and important task that has been undertaken with great effort. One of the greatest uncertainties of the SPMS technology is the lack of knowledge about the comparability of data sets, which emphasizes the importance of direct comparisons. This unique study is well designed and provides important insights into the performance, practical considerations and a fair and unbiased comparison of the different instruments and concepts. The manuscript is technically sound, interesting and well written; I have no concerns that this manuscript is suitable for publication in ACP.

We thank the reviewer for their time spent on the manuscript and their positive comments.

1. From their study, the authors gained detailed insights into some well-known general weaknesses of SPMS technology. In the conclusion/outlook, development directions for improvements and developments beyond the current technology could be discussed. For example, optical detection of all SPMS is limited to particles larger than ~150 nm (and not MS sensitivity) due to light scattering limitations. Strategies to improve quantification were also suggested (and referred to) and could be briefly discussed.

   Thank you for the suggestions. We have modified the corresponding contents in conclusions and outlook section (the 3$^{rd}$ and 2$^{nd}$ paragraphs from the end), as follows:

   Since the FIN-01 workshop, participants and other groups have been improving SPMS hardware as well as data analysis processes and particle type retrievals. For example, the ALABAMA has been implemented with a newly developed ADL system, a delayed ion extraction, and better electric shielding, resulting in higher DE for a wider size range and seven times higher intensities of the cation signals (Clemen et al., 2020). The next generation of the PALMS (PALMS-NG) has been updated with better particle sampling and optical design, which allow for the measurement of a wider size range (~100 nm to > 3 μm) and higher DE for the smaller particles (1 to 3 orders of magnitude improvement for the size < 200 nm), and a bipolar s-shaped mass spectrometer with higher mass resolution (can reach > 1000, formerly ~200) (Jacquot et al., 2024). A method for distinguishing K-feldspar from illite has been developed by Marsden et al. (2018) and subsequently used in analyzing ambient data (Marsden et al., 2019). Since not all the minerals have the same properties, e.g. ice nucleating ability (Atkinson et al., 2013), the capability to distinguish minerals is critical for the research field. Future investigations could consider improving our ability to further distinguish particles, minerals, and dusts specifically.

   Particle mass quantification, unaddressed in this paper, is another important contemporary topic. At the time of FIN01, SPMSs were qualitative. Since then, effort has been put into improving quantification. Froyd et al. (2019) is one example where particle type fractions measured by PALMS were propagated onto a size distribution obtained by a collocated optical particle spectrometer, thereby enabling the quantitation of particle number, surface area, volume, and mass concentrations. Such quantification techniques have been and can be used as a framework for other SPMSs. Since FIN01, SPMS measurements are now generally considered quantitative with uncertainties. A future workshop could focus on quantification.

2. L43/44: Repetitive wording "We found…"

   Thanks for pointing out this. We have modified the sentence as follows:

   We found that instrument-specific DE was more dependent on particle size than particle type, and particle identification favored the use of bipolar, rather than monopolar, instruments.

3. The abstract could also be a bit shorter. E.g. one of the studies key findings "We found that instrument-specific DE was more dependent on particle size than particle type." is somehow hidden behind detailed correlation data.

   We have modified the abstract by removing unnecessary sentences. However, we consider the "instrument-specific DE was more dependent on particle size than particle type" as an important finding. So, we still keep it.

   The revised abstract is now:

   Knowledge of chemical composition and mixing state of aerosols at a single particle level is critical for gaining insights into atmospheric processes. One common tool to make these measurements is single particle mass spectrometry. There remains a need to compare the performance of different single particle mass spectrometers (SPMSs). An intercomparison of SPMSs was conducted at the Aerosol Interaction and Dynamics in the Atmosphere (AIDA) chamber at the Karlsruhe Institute of Technology (KIT) in November 2014, as the first phase of the Fifth International Workshop on Ice Nucleation (FIN-01). In this paper we compare size distributions and mass spectra of atmospherically relevant particle types measured by five SPMSs. These include different minerals, desert and soil dusts, soot, bioaerosol (Snomax; protein granule), secondary organic aerosol (SOA) and SOA coated mineral particles. Most SPMSs reported similar vacuum aerodynamic diameter ($d_{va}$) within typical instrumental ranges from ~100–200 nm (lower limit) to ~2–3 μm (upper limit). In general, all SPMSs exhibited a wide dynamic range (up to ~$10^3$) and high signal to noise ratio (up to ~$10^4$) in mass spectra. Common spectral features with small diversities in mass spectra were found with high average Pearson's correlation coefficients, i.e., for average positive spectra $r_{avg-pos} = 0.74 \pm 0.12$ and average negative spectra $r_{avg-neg} = 0.67 \pm 0.22$. We found that instrument-specific DE was more dependent on particle size than particle type, and particle identification favored the use of bipolar, rather than monopolar, instruments. Particle classification from "blind experiments" showed that all instruments differentiated SOA, soot, and soil dust, and detected subtle changes in the particle internal mixing, but had difficulties differentiating among specific mineral types and dusts. This study helps to further understand the capabilities and limitations of the single particle mass spectrometry technique in general, as well as the specific instrument performance in characterizing atmospheric aerosol particles.

4. L89: 405 nm diodes ➔ 405 nm laser diodes

   Modified as suggested.

5. L99: The two step approaches do not only reduce fragmentation. As part of the ionization occurs in the particle's gaseous plume, quantification can be improved (Woods et al., 2001) and resonance effects can be used to increase the sensitivity to organics (Passig et al., 2022, Schade et al., 2019).

   Thank you for the explanation. We have added the statement as suggested.

   In addition, as part of the ionization occurs in the particle's gaseous plume, quantification can be improved (Woods et al., 2001) and resonance effects can be used to increase the sensitivity to organics (Passig et al., 2022; Schade et al., 2019).

6. L240 The LOD refers to the optical detection limit, so consider e.g. "…most of the particles were smaller than the optical detection limit of most SPMSs…."

   Thank you for making this clear. We have modified it as suggested.

7. L244: was the abbreviation DSF introduced before? Maybe I missed it…

   Yes, we have introduced DSF before this line. It is in equation (2). "…and $\chi_c$, $\chi_t$, and $\chi_v$ are the dynamic shape factors (DSFs) in the continuum, transition, and free molecular regime, respectively"

8.  L253: PALMS has high hit rates because the second detection laser is very close to the ion source. However, this design has also drawbacks, as solid-state ionization lasers cannot be used and the particle detection optic's layout is limited by the ion source. Therefore, ATOFMS and ALABAMA seem to be superior for ultra-fine particles, see e.g. Fig. 3c (Snomax).

    The reviewer is correct. While a solid laser can be used in the system if needed the choice of the excimer laser led to the separation distance. It is worth noting that we don't view either laser as better nor worse and that the laser choice leads to the separate distance (due to trigger time), not the other way around.

    We did not see evidence that separation distance, i.e., ATOFMS and ALABAMA, are superior for ultra-fine particles (smaller than 100 nm aerodynamic diameter, $d_a$). Their DEs for the Snomax in the size range of 200 to 350 nm vacuum aerodynamic diameter ($d_{va}$, similar as $d_a$) were comparable with that of PALMS (refer to Fig. 3c in the revised manuscript).

9.  L263: Could the DE values for both modes of the miniSPLAT be shown in table 1?

    Since the terminology in DE in the miniSPLAT reference is different from the other SPMSs and the DE values for PSL particles for both modes are not provided, we would like to leave as it is.

10. L330: Two Typos: "ALABMA" and "LAATOF"

    Both revised.

11. Table 1: The ATOFMS sizing lasers have probably 532 nm wavelength, if they are Nd:YAG.

    Thank you for this point. For ATOFMS, the lasers used in FIN01 were two laser diodes (Livermore Instrument Inc.; 405 nm), rather than Nd:YAG. We have revised this in Table 1.

12. Table 1: Use mm or cm consistently.

    Revised, using only mm for consistency.

13. Table 1: Hit Rate PSL for miniSPLAT should also be annotated with footnote "b".

    We have annotated both HR and DE for miniSPLAT with footnote "b".

14. Table 3: "One fork shape" and "Two fork shape" sound nice but should be put in quotation marks.

    Both are now quoted.

15. Fig 1: In case you have problems with the length of the manuscript, consider to remove figure 1 as it contains no important information for the reader.

    We have moved figure 1 to the SI (now figure S1).

**References**

Woods, E.; Smith, G. D.; Dessiaterik, Y.; Baer, T.; Miller, R. E. *Anal. Chem.* **2001**, *73*, 2317– 2322

J. Passig, J. Schade , R. Irsig , T. Kröger-Badge , H. Czech , T. Adam , H. Fallgren , J. Moldanova , M. Sklorz , T. Streibel and R. Zimmermann, *Atmos. Chem. Phys.*, 2022, **22** , 1495 —1514

J. Schade , J. Passig , R. Irsig , S. Ehlert , M. Sklorz , T. Adam , C. Li , Y. Rudich and R. Zimmermann, *Anal. Chem.*, 2019, **91** , 10282 —10288

**Referee #2 comments:**

The manuscript describes the intercomparison and blind testing of five Single Particle Mass Spectrometers (SPMS) that were operated in parallel at the AIDA aerosol chamber. A number of different aerosol types were supplied to the chamber and size distribution measurements as well as mass spectra for the SPMSs were compared. Correlations between the instruments and between the different particle types are discussed. Two blind experiments were conducted where a mixture of three particle types (SOA, soil dust and graphite soot) were supplied to the chamber, but the participants of the blind experiments did not know about the particle types, sizes and relative fractions.

The intercomparison constitutes an important study to evaluate the performance and limitations of SPMS instruments. Although the instruments are demonstrated to be generally capable of identify and classify different aerosol types, the (quantitative) results can vary widely (e.g., Fig. 10, P1, ALABAMA suggests >90% pure SOA particles, while ATOFMS detects Illite NX and K-feldspar in >90% of the particles).

We thank the reviewer for their time spent on the manuscript and their positive comments.

My main comment concerns the discussion of the blind tests. This discussion should be expanded.
1.  The authors should add the pie charts of the mixture introduced into the chamber (already shown as inserts in Fig. 2) again in Fig 10.

Thank you for the suggestion. We have added those pie charts as f1) and f2) in the Fig 10 (current Fig. 9)

[Figure]

Figure 9: Particle classes and their relative contributions for blind periods P1 and P2 for: a1-2) PALMS, b1-2) ALABAMA, c1-2) LAAPTOF, d1-2) ATOFMS and e) miniSPLAT (which only reported data in P1). f1-2) are the same pie charts as those in Fig. 1 added here for better comparison. The plots shown here are the data provided to the referees after experiments (i.e., before participants knew the composition of the blind experiment). Note the left pie chart for miniSPLAT represents acquired mass spectra, consistent with what is presented for the other SPMSs. The right pie chart, provided to the referees after experiments, represents calibrated data (see text for details). Particle clustering is shown for positive and negative spectra separately for the unipolar switchable PALMS instrument.

2. It is noteworthy that the miniSPLAT with calibration is able to accurately reproduce the original particle mix. The discussion of the calibration procedures for miniSPLAT should be expanded and/or a reference for the calibration procedures should be given. Can such a calibration be regularly achieved, e.g., for field measurements? Furthermore, it should be discussed whether such calibration would also be possible for the other instruments.

   The miniSPLAT DE calibrations were determined in a series of laboratory experiments using many particle types, including SOA, soot, and dust (Vaden et al., 2011). Such calibrations take into account particle beam divergence, which depends on particle composition/shape/morphology, and size. Aspherical particles form particle beams with higher beam divergence, resulting in a decrease in DE. Particle beam divergence is measured by miniSPLAT with temporal resolution of 1 sec (Vaden et al., 2011). Its relationship to other particle properties was demonstrated in both, laboratory and fields studies (Vaden et al., 2011). Such calibration is also possible for the other SPMSs, not limited to the instruments used in FIN01.

   We have modified the corresponding sentences in the 2nd paragraph of section 3.2.3, as follows:

   The right pie chart, provided to the referees, uses the DE calibrations, determined by miniSPLAT for many particle types, including SOA, soot, and dust (Vaden et al., 2011). Such calibrations take into account particle beam divergence, which depends on particle composition, shape, and morphology. The applied DE calibrations yield the 2nd pie chart (Fig. 9 e right) which can be compared to the original particle mix (Fig. 1 and Fig.9 f1). Note that the other instrument groups did not produce a similar calibrated pie chart in this study, however, this can be achieved if similar calibration for multiple particle types was done by using the other SPMSs. Such calibration can also be achieved in the field but with larger uncertainties due to the chemical and morphology complexity of the ambient particles.

3. It should be discussed to what extent measurements of the individual SPMS instruments are considered to be not quantitative at all, or semi-quantitative, or quantitative after calibration. Can uncertainties be provided for each instrument?

   Thank you for your suggestion. This is similar to the first point of referee #1, please see that review and our response for more information.

4. A strong caveat should be added that some seemingly obvious interpretations of such pie charts from SPMS results are not possible, and that such pie-charts can potentially be quite misleading (e.g., ALABAMA classifying >90% of the particles as SOA does not mean that the aerosol is actually dominated by SOA; four instruments seeing less than 5% soot particles does not mean that soot is only a minor component of the aerosol, etc.).

   We have expanded this part of the paper with the added text per the Reviewer comment:

   It is noteworthy that caution should be utilized in the interpretations of the presented pie charts. For example, all the instruments classified the majority of particles as SOA and identified less than 10% soot particles. These results were in the context of certain size range, i.e., ~100–200 nm to ~2–3 μm $d_{va}$. To obtain more accurate number fractions, composition and size-dependent DE need to be considered.

Overall, the manuscript is well written and clear. The manuscript is clearly suitable for ACP after considering my main comment. The comprehensive intercomparison of the performance of the individual SPMS instruments is very insightful and valuable for interpreting SPMS results in general and for understanding the current limitations of each instrument.

We thank the reviewer again for the positive comments.

Technical correction:

5. Line 857: "Note that…"

Thank you for pointing it out. We have revised.

**Responses to Referees**

**References**

Vaden, T. D., Imre, D., Beránek, J., and Zelenyuk, A.: Extending the Capabilities of Single Particle Mass Spectrometry: I. Measurements of Aerosol Number Concentration, Size Distribution, and Asphericity, Aerosol Science and Technology, 45, 113–124, https://doi.org/10.1080/02786826.2010.526155, 2011.